# LLM-as-a-Prophet: Understanding Predictive Intelligence with Prophet Arena 🔮 [*]

**Qingchuan Yang** [1][†], **Simon Mahns** [2][†], **Sida Li** [3][†], **Anri Gu** [3][†], **Jibang Wu** [4][‡], **Haifeng Xu** [3][‡]

[1] University of Southern California  [2] Meta  [3] The University of Chicago  [4] New York University

## Abstract

With the rapid progress of large language models (LLMs) trained on every available piece of data, it becomes increasingly challenging to reliably evaluate their intelligence due to potential data contamination and benchmark overfitting. To overcome these challenges, we investigate a new angle of benchmarking LLMs' intelligence by evaluating their capability in forecasting real-world future events, a paradigm we call "LLM-as-a-Prophet". Such forecasting tasks require combination of sophisticated capabilities while remaining free from data contamination or overfitting. To systematically evaluate such predictive intelligence of LLMs, we introduce `Prophet Arena`, a general evaluation benchmark that continuously collects live forecasting tasks and decomposes each task into distinct pipeline stages, supporting our controlled and large-scale experimentation. Our comprehensive evaluation reveals that many LLMs already exhibit impressive forecasting capabilities, reflected in, e.g., their small calibration errors, consistent prediction confidence and promising market returns. However, we also uncover key bottlenecks even in frontier models, such as inaccurate event recalls, misunderstanding of data sources and slower information aggregation compared to markets when resolution nears.

## 1 Introduction

Forecasting is a fundamental intellectual pursuit that has shaped human progress from the earliest scientific inquiries to modern economics and finance. In machine learning, forecasting has also been a central theme, with rich traditions ranging from time-series analysis (Chatfield, 2000) and online learning (Foster & Vohra, 1998) to conformal prediction (Barber et al., 2023). Yet, somewhat surprisingly, the challenge of open-domain forecasting, producing accurate predictions across a wide range of topics without domain-specific tuning or specialized datasets, remains largely unexplored. Achieving reliable foresight in this setting would represent a qualitative leap in AI capability, with far-reaching societal implications, from enhancing market efficiency to guiding high-stakes policy decisions (Arrow et al., 2008).

At its core, forecasting is the process of connecting present knowledge to anticipate future outcomes. Large language models (LLMs) seem natural candidates for this role. Trained on massive corpora of human knowledge through the seemingly narrow objective of next-word prediction, LLMs have developed emergent capabilities that extend far beyond their training objective (Bubeck et al., 2023). This motivates the prospect that the ability to predict the next word may also give rise to the ability to predict the next event. Indeed, recent works by Zou et al. (2022); Halawi et al. (2024) have already made promising progress in employing language models for forecasting tasks. We envision that, with growing interest and continued progress, this would position LLMs not only as repositories of human knowledge but also as instruments of reliable foresight, leading to the prospect of **LLM-as-a-Prophet**: *Can AI systems reliably predict the future by connecting the dots across existing real-world information?*

In this paper, we seek to systematically examine the prospects and challenges of building general-purpose systems for open-domain forecasting. On one hand, forecasting is a natural next pursuit

---

[*]Correspondence to `contact@prophetarena.co`.
Part of the work by Qingchuan Yang and Jibang Wu are done while they are at the University of Chicago. We thank Alex Gu, Chaplin Huang and Lucien Liu for help at the early stage of this work.

[†]Equal contribution.

[‡]Equal advising.

given the rapid progress of AI, as it draws on a combination of advanced capabilities that current models are only beginning to demonstrate: information retrieval, complex reasoning and data analysis. Moreover, as many established benchmarks are approaching saturation and are increasingly prone to training-data contamination (Deng et al., 2024), open-domain forecasting provides a forward-looking and contamination-free setting with objectively measurable outcomes, making it a rigorous testbed for evaluating advanced model intelligence. On the other hand, we observe that current LLMs often struggle with key requirements for reliable foresight, including calibrated uncertainty estimation (Geng et al., 2023) and robust reasoning (Zhou et al., 2024) in the presence of noisy or incomplete evidence. As a result, their forecasting results may at times resemble guesswork rather than deliberated prediction, raising the possibility that fundamental barriers must be addressed before such evaluations can serve as a meaningful benchmark at the present time (Paleka et al., 2025a). Toward this end, we introduce `Prophet Arena`, a general framework for evaluating LLMs on live, real-world forecasting questions in a controlled and extensible way. Our goal is not only to assess the current forecasting performance of LLMs, but also to use forecasting as a lens for studying core components of intelligence, including reasoning, calibration, evidence aggregation. By doing so, we aim to identify which capabilities are emerging, which remain limited, and how forecasting evaluation can guide the development of more reliable predictive intelligence.

**Our contributions.** First, we introduce the paradigm of LLM-as-a-Prophet, framing real-world open-domain forecasting as a rigorous and forward-looking measure of intelligence. This perspective highlights forecasting as not only an application domain but also a unifying task that brings together many advanced capabilities, making it a natural benchmark for the next phase of AI development. Second, we present `Prophet Arena`, a live and extensible benchmark that continuously collects real-world forecasting questions across diverse domains. The framework decomposes forecasting into distinct stages for controlled evaluation and incorporates multiple scoring metrics, including statistical accuracy, calibration, and economic value, to provide a comprehensive view of predictive quality. Third, we conduct large-scale experiments on leading LLMs using `Prophet Arena`. Our evaluation uncovers areas where models demonstrate non-trivial foresight, while also identifying systematic weaknesses – particularly in interpretation of data sources, bottlenecks in reasoning traces, and conservativeness in predictions.

## 1.1 CONNECTION AND COMPARISON TO PREVIOUS WORKS

We briefly highlight our work's connection and comparison to previous works, while defer more in-depth discussions to Appendix A.

**Understanding and advancing LLMs' forecasting capabilities.** A key goal of our work is to understand the *LLM-as-a-Prophet* paradigm by measuring LLMs' current forecasting capabilities and analyzing how other capabilities like knowledge internalization and source usage shape predictive intelligence. This connects to recent studies examining *specific aspects* of LLM forecasting, such as *temporal generalization* (Dai et al., 2025; Zhu et al., 2025) and *forecast consistency* (Paleka et al., 2025b). Like these works, we build a benchmark platform `Prophet Arena`, but ours is, to our knowledge, the first to diagnose the *general* predictive intelligence of LLMs. We also note recent progress on *improving* LLM forecasting capabilities (Zou et al., 2022; Halawi et al., 2024). While our focus is benchmarking rather than model design, our insights about LLM strengths and limitations may inform future advances in forecasting.

**Forecasting benchmarks.** Forecasting has recently become a widely used LLM benchmark since it challenges multiple capabilities while avoiding data contamination by testing models on future events (Dai et al., 2025; Karger et al., 2025). To our knowledge, one of the earliest effort is ForecastQA (Jin et al., 2021), which tests AI models on crowdsourced multiple-choice forecasting questions. Subsequent work created increasingly difficult benchmarks by incorporating prediction-market events (Zou et al., 2022), extracting events from news (Zhang et al., 2024; Wang et al., 2025), curating future-oriented questions from websites (Wildman et al., 2025), using open-ended forecasting prompts (Guan et al., 2024), and constructing dynamic, live-updating benchmarks (Zeng et al., 2025; Karger et al., 2025; Bianchi et al., 2025).

We also develop a benchmark, `Prophet Arena`, which we intentionally ground on thousands of prediction markets events, leveraging their incentive-aligned participation and standardized resolution criteria to obtain a more rigorous testbed that mitigates selection bias from question design. Beyond ranking models, we are also interested in what conclusions we can draw from the analysis of the models' performance. Accordingly, we provide a comprehensive evaluation of LLMs'

predictive intelligence, including Brier scores, calibration and economic value – and we thoroughly discuss how they capture different aspects of a forecast in Section 3.1. In contrast, previous benchmarks mostly focuses on one metric, such as Brier score (Halawi et al., 2024; Karger et al., 2025), calibration (Zou et al., 2022) and accuracy (Dai et al., 2025; Zhang et al., 2024). Moreover, beyond evaluation metrics, we analyze how specific LLM capabilities (e.g., internalization, source usage) shape forecasting behavior.

## 2 PROPHET ARENA: A LIVE BENCHMARK FOR PREDICTIVE INTELLIGENCE

`Prophet Arena` is implemented as a live, continuously updated pipeline for evaluating forecasts on real-world events. As illustrated in Fig. 1, it consists of three main stages, which together form an end-to-end workflow for assessing predictive intelligence at scale. Below we describe the design of each stage at a high level and defer the implementation details to Appendix B.

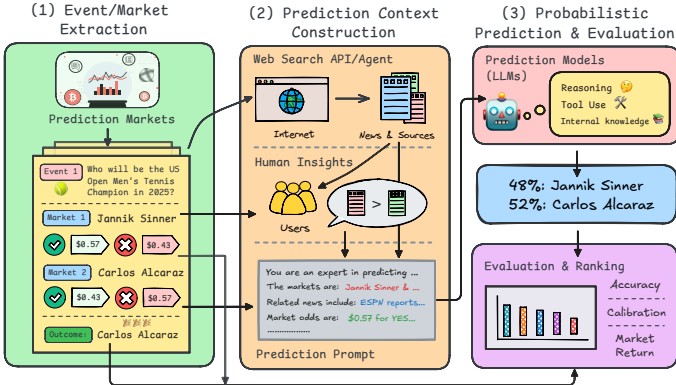

Figure 1: **Flowchart for the full `Prophet Arena` pipeline.**

**(1) Event and Market Extraction.** `Prophet Arena` collects unresolved events from Kalshi, a live prediction platform with events spanning across categories such as finance, sports, politics and entertainment. Kalshi provides structured, real-time events with verifiable outcomes, enabling standardized and comparable evaluation across models. To ensure informativeness and comparability, we filter by *Popularity* (volume/liquidity/volatility), *Diversity* (domain balance), and *Recurrence* (repeated formats). `Prophet Arena` periodically retrieves a number of unresolved events, making it a truly live benchmark, rather than a fixed, static dataset susceptible to training data contamination. The use of prediction markets guarantees that all questions pertain to genuine future outcomes grounded on real world importance and that their resolutions can be objectively verified once finalized.

**(2) Prediction Context Construction.** For each event, `Prophet Arena` constructs a unified *prediction context* that all models receive identically. This context contains: (1) Relevant information sources retrieved by an LLM-based search agent using web queries that collect titles, snippets, timestamps, and URLs of recent news or reports; and (2) Market snapshots including the latest `Yes/No` contract prices and trading volumes, from which implied probabilities (based on market contract prices) are derived. Providing identical contexts isolates differences in models reasoning and calibration rather than their retrieval capabilities. The search component in `Prophet Arena` is fully *searcher-agnostic*: new search agents can be added or replaced without altering the forecasting protocol or evaluation procedure. In all experiments presented in this paper, we use a single LLM-based searcher instantiated with GPT-4o equipped with web access.

**(3) Probabilistic Forecasting and Comprehensive Evaluation.** Given an event and its constructed context, each model produces a probabilistic forecast for every market within the event. The model outputs a predicted probability, interpreted as its belief that the market will resolve to `Yes`, together with a short natural-language rationale. All forecasts are logged for later analysis but only the probabilities are used for quantitative evaluation. Once the event resolves and the true outcomes become available, `Prophet Arena` evaluates the forecasts along multiple complementary metrics – which are detailed in Section 3. Together with the event extraction and context construction stages, this final step completes the `Prophet Arena` pipeline – turning live, real-world questions into a continuous, contamination-resistant benchmark for predictive intelligence.

**(*) Bonus feature: Market Baseline.** To establish a fair and interpretable anchor for comparing different LLM forecasters, we include a *Market Baseline* – a synthetic forecaster whose prediction is defined as the market-consensus probability that market will resolve to `Yes`. In practice, this probability is inferred from the normalized contract prices. For instance, if the `Yes` and `No` contract prices are 0.8 and 0.2, respectively, then the market baseline assigns an 80% probability to the `Yes` outcome.[1] As we will demonstrate in Section 3, this market baseline serves as an informative benchmark for assessing *forecast difficulty* and contextualizing model performance. When LLMs outperform the market baseline, they demonstrate genuine predictive advantage over the aggregated human consensus in the real-world market.

**On the Design Choices of `Prophet Arena`.** Despite the few recent benchmarks focused on forecasting, `Prophet Arena` is distinguished by the following key design choices (see Table 4), all of which are tailored for our systematic evaluation of LLM's forecasting capabilities. To our best knowledge, `Prophet Arena` is the first large-scale benchmark that continuously runs evaluation on real-market events under a multi-horizon protocol and within a modularized forecasting pipeline.

1. **Probabilistic forecasts:** Future events are intrinsically random, so we elicit each market's probabilistic forecast from each model, rather than a single choice of the most likely outcome. Notably, probabilistic forecasting is also the standard in real-world forecasting platforms, such as Metaculus (2015) and Good-Judgement-Open (2015), as well as in the design of forecasting systems (Halawi et al., 2024).

2. **Multi-horizon protocol:** We implement a multi-horizon forecasting protocol that sets a schedule for the models to make prediction across various timestamps before the event resolves; we defer details on the scheduling algorithm and multi-horizon aggregation to Appendix B.3. This enables our temporal analyses of how models update their forecasts as market conditions and public information evolve across the full lifecycle of an event.

3. **Modularized forecasting pipeline:** The process of prediction is broken down from source collection to probability elicitation to market actionability. This enables a comprehensive and controlled study of LLMs' forecasting performance at different modules.

4. **Market return metrics:** `Prophet Arena` allows the evaluation of market profitability of each forecast, measuring models' *relative* advantage over market consensus.

| Benchmark | Live events | Probabilistic | Multi-horizon | Modularized | Return metrics |
|---|---|---|---|---|---|
| MIRAI | – | – | ✓ | – | – |
| FORECASTBENCH | ✓ | ✓ | ✓ | – | – |
| FUTUREBENCH | ✓ | – | – | – | – |
| FUTUREX | ✓ | – | – | – | – |
| **Prophet Arena** | ✓ | ✓ | ✓ | ✓ | ✓ |

Table 1: **Comparisons with related forecasting benchmarks**, including MIRAI (Ye et al., 2024), ForecastBench (Karger et al., 2025), FutureBench (Bianchi et al., 2025), FutureX (Zeng et al., 2025).

## 3 EVALUATION METRICS AND EMPIRICAL RESULTS

To obtain a thorough evaluation, we consider a number of different evaluation metrics that measure various aspects of LLMs' forecasting. Specifically, we consider three metrics: (1) *Brier scores* (Brier, 1950; Gneiting & Katzfuss, 2014) which measures the *absolute* quality of a probabilistic prediction; (2) *calibration error* (Murphy, 1973; DeGroot & Fienberg, 1983; Guo et al., 2017) which measures the statistical consistency of a probabilistic prediction; and (3) *market return* (Mallikarjuna & Rao, 2019) which measures the *relative* advantage over current market's consensus.[2] We note that recent works on forecast evaluations have considered Brier scores (e.g., (Halawi et al., 2024; Karger et al., 2025)) and calibration errors (e.g., (Zou et al., 2022)), though no work has systematically evaluated models' market returns to our knowledge.

---

[1]In practice, contract prices may not sum up to one due to exchange fees, requiring slight normalization.

[2]Market consensus about an event's forecast is generally difficult to obtain. However, a good proxy of such data is available for `Prophet Arena` as it fetches forecasting events from prediction markets, which are widely believed to offer good approximation of the market's consensus on the event's probability (Arrow et al., 2008; Berg et al., 2008).

Let $E_i$ denote the $i$-th event ($i = 1, \ldots, n$), and $M_{ij}$ the $j$-th market within $E_i$ ($j = 1, \ldots, m_i$), where $m_i$ is the number of markets for event $E_i$. For each market, a model outputs a predicted probability $p_{ij} \in [0, 1]$ of resolving Yes, and the realized outcome is $o_{ij} \in \{0, 1\}$. A complete notation table with full definitions is provided in Appendix B.

## 3.1 THREE EVALUATION METRICS AND THEIR DIFFERENCES

In this subsection, we first describe the three evaluation metrics we employ, followed by an illustration of their differences and what they are used for.

Proper scoring rules (Gneiting & Raftery, 2007) directly evaluate the quality of the predicted probabilities. Prophet Arena adopts a classic and strictly proper scoring rule, the **Brier score** (Brier, 1950), defined for event $E_i$ with markets $\{M_{ij}\}_j$ as $\text{BS}_i = \frac{1}{m_i} \sum_{j=1}^{m_i} (p_{ij} - o_{ij})^2$. That is, the Brier score for an event $E_i$ is the squared distance between the forecasted probability and indicator of the event realization, averaged across the markets.

Our second metric is the **expected calibration error (ECE)** (DeGroot & Fienberg, 1983; Guo et al., 2017; Kalai & Vempala, 2024), which captures the following discrepancy of a forecaster – given that it predicts probability $\tilde{p}$ to Yes, how different the predicted $\tilde{p}$ is from Yes's real occurrence frequency. The formal definition of ECE and its empirical estimation are standard in the literature, hence we defer its detail to Appendix C.2. ECE is also classically known as the *reliability* score of a forecaster (Murphy, 1973; Guo et al., 2017). Lower ECE means the forecast is more reliable, in the sense that the predicted probabilities are closer to the (conditional) true probabilities.

Finally, since Prophet Arena uses events from real-world prediction markets, it allows us to evaluate the economic value of a forecast based on current market prices. Our third metric is thus **Average Return**, capturing *how profitable it would be to trade in the market using LLM forecasts?* Prophet Arena allows the evaluation of the average return under optimal strategies targeting different risk levels[3], but in the main body we report the return of an optimal strategy under *risk-neutrality*, averaged across all markets with a unit budget allocated to each market.

**Differences among the Three Metrics and What Each Metric is For.**    Differences among the three metrics are already reflected in our empirical evaluations below. However, it is still conceptually useful to understand how these metrics fundamentally differ and what each metric is for. First, the difference between the Brier score and ECE is well-known (Murphy, 1973).[4] A forecast with better/smaller ECE could have worse Brier score (see an example in Appendix C.4). In practice, ECE is crucial when decision makers have *risk preferences* since smaller ECE ensures that the risk encoded in the forecasted probabilities is more reliably captured, leading to better risk-preference-adjusted decisions – even when the forecasted probabilities have worse Brier scores. In Appendix C.4, we illustrate this insight with a simple example in which a forecaster with better ECE but much worse Brier score could lead to better risk-adjusted utility.   Second, while both market return and Brier score assess forecast quality, they differ fundamentally. The Brier score is an *absolute* metric, measuring a forecasts closeness to the ground truth, independent of external factors like market prices. In contrast, market return is a *relative* metric, capturing how much a forecast outperforms the markets current belief (interpreted as the contract price (Wolfers & Zitzewitz, 2006)). This distinction is evident in our evaluations, and Appendix C.5 provides an illustrative example where forecasts with worse Brier scores achieve higher market returns. Finally, while calibration is not related to the total market return, but is indicative about how balanced the market return is from betting on the Yes contracts and No contracts. To formalize this intuition, we prove in Appendix C.6 that a well-calibrated and symmetric forecaster – intuitively, one that is not systematically more aggressive or conservative than the market – will have balanced expected returns from both contract types.

## 3.2 PERFORMANCES ACROSS DIFFERENT DIMENSIONS AND DISCUSSIONS

Throughout the main body of the paper, we highlight (the same) five representative LLMs[5] out of the 22 evaluated in total (the full results are available in Table 7 of Appendix D.1). As shown in

---

[3]Appendix C.3 offers derivations of such strategies using risk-sensitive utility functions from classic behavioral economics.

[4]Murphy (1973) shows that the Brier score can be decomposed into three terms: the ECE (also coined it "reliability score" by Murphy (1973)), the *uncertainty* score that captures inherent randomness of the to-be-forecasted event, and the *resolution* score that captures forecasts' variance.

[5]Models are chosen to span proprietary and open-source families, reasoning and non-reasoning variants, and a range of performance levels in the full ranking.

| LLM | Forecasting Loss | | Calibration Error | | Market Return | |
|---|---|---|---|---|---|---|
| | ↓ **Brier** (95% CI) | **Rank** | ↓ **ECE** | **Rank** | ↑ **Average** (95% CI) | **Rank** |
| GPT-5ᴿ △ | 0.184 (± 0.006) | ① | 0.042 | ② | 0.943 (± 0.042) | ① |
| Grok 4ᴿ △ | 0.189 (± 0.005) | ② | 0.043 | ③ | 0.864 (± 0.052) | ④ |
| Claude Sonnet 4ᴿ △ | 0.194 (± 0.006) | ③ | 0.041 | ① | 0.909 (± 0.101) | ② |
| Gemini 2.5 Flashᴿ △ | 0.197 (± 0.007) | ④ | 0.067 | ⑤ | 0.883 (± 0.053) | ③ |
| Llama 4 Scout △ | 0.219 (± 0.008) | ⑤ | 0.060 | ④ | 0.805 (± 0.040) | ⑤ |
| Market Baseline | 0.187 (± 0.006) | N/A | 0.069 | N/A | 0.899 (± 0.043) | N/A |

Table 2: **Evaluation of five representative LLMs.** For Brier and Average Return, bootstrapped 95% confidence intervals are reported. Superscript ᴿ is used to denote a reasoning model, with its reasoning configuration in Appendix B.6. The full results for all 23 LLMs are provided in Table 7. Although our benchmark updates in real time, for the purpose of writing this paper we need to fix a dataset. Our evaluation is conducted on 1,367 events that were resolved before October 11, 2025.

Table 2, frontier proprietary models consistently outperform the *Market Baseline* across all metrics. Here, the *Market Baseline* refers to a "synthetic" forecaster that treats market prices as its own predicted probabilities (Wolfers & Zitzewitz, 2006): specifically, for each market, it uses the normalized market price for Yes contract as its probabilistic forecast that the market will happen.

Notably, the relative rankings of models differ depending on which evaluation metric is used, illustrating the complementary perspectives offered by accuracy, calibration, and profitability. Concretely, Brier scores fall in a narrow band $[0.17, 0.24]$ (pure random guess has expected Brier score $\sim 0.25$). By contrast, calibration differences are more pronounced: strong models typically achieve ECE $\leq 0.05$, whereas weaker ones fall in the $[0.05, 0.2]$ range. For market performance, even GPT-5ᴿ, the top-ranked model, fails to reach break-even (Average Return $< 1$), and most models fall below 0.9. Since event-level payoffs depend heavily on market-implied probabilities, the resulting returns exhibit substantial variance, as evidenced by the wide confidence intervals. In Appendix D.2, we further discuss the *Sharpe ratio* (Sharpe, 1998) of our betting strategy, which normalizes Average Return by volatility, providing a more stable comparison of models' economic performance.

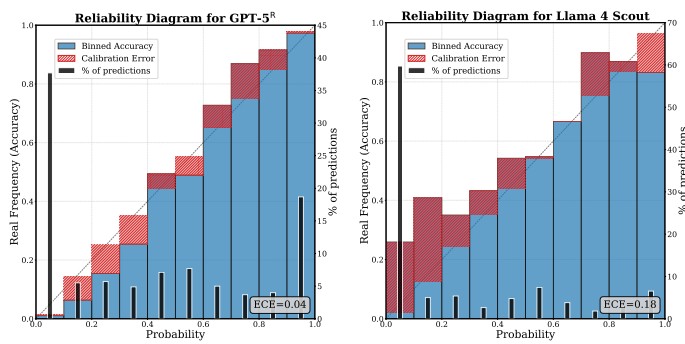

Figure 2: **Reliability diagrams for the best and worst LLMs ranked by calibration score (ECE).** The **black** histogram indicates fraction of predicted probabilities in each bin. The calibration error within each bin appears as the height of the red rectangle, i.e., the gap between accuracy and confidence. The reported ECE score (bottom right) corresponds to a weighted sum of these errors, weighted by the distribution. Superscript ᴿ denotes a reasoning model.

Overall, our results suggest that absolute forecasting skill and relative profitability against prediction markets are still challenging for todays LLMs. A more fine-grained investigation could help deepen our understanding about what could make an LLM a good prophet. Fig. 2 draws for the best and worst models (Left & right) their reliability diagrams regarding calibration errors (Guo et al., 2025), where predicted probability ($x$-axis) is compared against realized frequency ($y$-axis) at different probability bins. While their calibration is similar in intermediate ranges, the stronger model – GPT-5 – performs much better in the extreme bins (0-0.1 and 0.9-1.0), where it almost always predicts correctly. Because such extreme forecasts occur frequently, this advantage helps explain the gap in both Brier score and market return.

## 4 AN IN-DEPTH EVALUATION OF LLM-AS-A-PROPHET

In this section, we employ our customized design of `Prophet Arena` to conduct a systematic suite of experiments that shed light on the prospects of LLM-as-a-Prophet.[6] The events used for evaluation were collected via our real-time Kalshi data pipeline, and consequently the distribution of event categories reflects the underlying composition: 81% Sports, 5% Entertainment, 5% Politics, and 9% Other. To ensure that the results remain robust to event composition, we additionally verify that these performance patterns are robust to reweighting the dataset toward a more balanced distribution across categories. Full results from these balanced-subset evaluations are reported in Appendix C.8.

### 4.1 MECHANISTIC ANALYSIS OF LLM-AS-A-PROPHET

From a mechanistic perspective, we seek to understand how forecasting is shaped by the interplay of distinct capabilities, and in doing so, pinpoint the bottlenecks and causal links that determine their effectiveness. To this end, we design experiments that examine the key factors behind strong forecasting performance, ranging from a model's internal knowledge, to the quality and accessibility of external sources, to its ability to integrate those sources effectively.

#### 4.1.1 CAN INTERNALIZED KNOWLEDGE BECOME FORESIGHT?

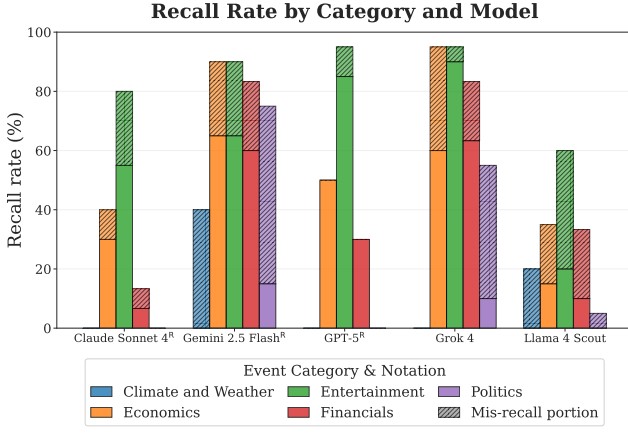

Figure 3: **Recall rate by event category and model.** Computed using the knowledge-internalization recall prompt (Appendix F.3.1). The shaded region indicates the fraction of events the model claims to recognize but recalls incorrectly.

Predictive intelligence is a complex decision-making process which relies not only on information retrieval, but also on *internalized* knowledge. One concrete question is to ask whether models are merely reproducing surface-level details or leveraging deeper knowledge of past outcomes to inform present forecasts. This requires us to study how models interpret and recall events. Hence, we retrieve 100 past events from Kalshi before the models' knowledge cutoff dates, and evaluate models under a recall prompt in Appendix D.8.2.

**Event recall varies by topics and models.** In Fig. 3, we observe that models most reliably recall events in *Entertainment*. By contrast, *Climate and Weather* and *Politics* display low recall and frequent mis-recall. Two factors likely contribute. First, *Weather* prompts often require fine-grained, date-stamped facts (e.g., *Highest temperature in Miami on Aug 29, 2023?*), which are less likely to be memorized. Second, due to 2023 regulatory constraints on Kalshi's election markets, *Politics* in our dataset skews toward politics-adjacent indicators (e.g., "*Biden 538 approval rating on Aug 30, 2023?*"), which varies daily and may be sparsely represented in training corpora (Kalshi (2025)).

Despite these broad patterns, models differ in recall accuracy. For *Economics* and *Politics*, GPT-5 (High) correctly answered all events it claimed to recall. In contrast, models like Llama 4 Scout and Gemini 2.5 Flash (Reasoning) reported recognizing events in all categories. However, for categories like *Climate and Weather* as well as *Politics*, almost all of their recalled events were false.

**Event recall is approximate, not precise.** Consider the event "*Billboard Hot 100 #1, Jul 13, 2023?*". Gemini 2.5 Flash recalled (Appendix E.2) that Olivia Rodrigo's *Vampire* displaced Morgan

---

[6]Due to the resource constraint, our experiment results in this section uses a subset of 100 events sampled uniformly from our full benchmark. This dataset is publicly available at `https://huggingface.co/datasets/[redacted-for-anonymity]`.

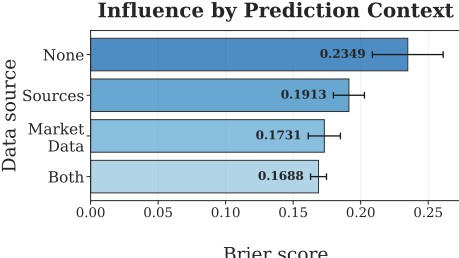
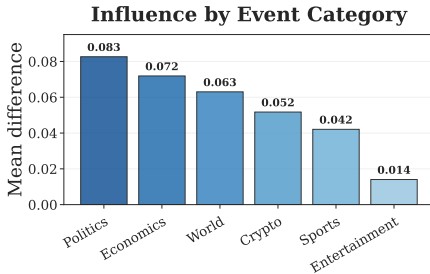

Figure 4: **Prediction quality across different contexts and event categories.** Left panel shows average Brier scores across evaluated LLMs for prompts with varying information availability (none, sources only, market data only, or both). Error bars represent the interquartile range (IQR) of model performance. Right panel displays mean Brier scores of events in each category.

Wallen's *Last Night* at number one, which is correct. However, it aligned the answer with the chart dated July 15, 2023, treating it as equivalent to July 13, 2023. Thus, while the model demonstrated knowledge of the outcome, it failed to recall the precise alignment between dates and chart releases. This illustrates that event recall in LLMs is approximate: models often retain coarse associations (the correct song and transition) but lack fidelity on exact temporal details.

### 4.1.2 HOW DO CONTEXTS SHAPE FORECASTS?

Given that the `Prophet Arena` prediction inputs consist of existing prediction market data alongside multiple relevant news sources, we analyze how these different information sources affect model performance. Fig. 4 shows the average Brier scores across all tested LLMs under four conditions: access to both market data and news sources, market data only, news sources only, and none. The results demonstrate a clear performance hierarchy; as expected, models with access to both market data and news sources output the best predictions, while those without access to either source of information exhibit the poorest performance.

Beyond performance differences, the variance patterns reveal deeper insights. Interestingly, models using only market data perform only slightly worse, on average, than those with both market data and news sources; however, the key difference lies in the variability of predictions. Combining multiple high-quality sources substantially reduces variance in prediction quality, suggesting that sources still offer valuable signals and perspectives that enable more consistent forecasting. Thus, while market data is powerful precisely because it aggregates information from a plethora of sources and trends, adding a few carefully chosen high-quality sources can still help stabilize and refine the signals the information provides. Appendix D.9 includes a deeper analysis of LLMs abilities to find and utilize high-quality sources.

**Sources can clarify or confound predictions.** As shown in Fig. 4, on average, adding sources improves mean Brier score. However, the effect is heterogeneous: not only does the magnitude of the effect vary based on event category, it also does not necessarily strictly improve prediction quality, as shown in the case study in Appendix E.3. Regarding category differences, the benefit of adding sources is not uniform. In areas like politics, where events can be interpreted through multiple perspectives, incorporating information from varied outlets and institutions appears to add useful context. In contrast, in domains such as entertainment or sports, the marginal value of additional sources seems smaller. This highlights that more information is not necessarily better; the effectiveness of sources depends on their relevance to the prediction task.

### 4.1.3 MODELS ARE MORE CONSERVATIVE THAN MARKETS

Fig. 5 compares model prediction probabilities to the market-implied probabilities. Since market data is included in the LLM prompts – most models generally align closely with market predictions. However, across a large majority of events, LLMs consistently output more conservative probabilities. The clearest example is Llama 4 Scout, shown in Fig. 5d: even when markets assign near-certain probabilities (close to one), the model remains hesitant, rarely producing equally extreme predictions. This reflects a systematic conservatism, where the model underweights outcomes the market views as almost certain. As illustrated in Fig. 5a & Fig. 5b, although both GPT-5 and Grok 4 adopt a cautious approach in their probability assignments, they generally track the market more closely across the full probability range and avoid the same degree of reluctance. Claude Sonnet 4

exhibits a slightly more assertive pattern: in the mid-probability range (around 0.5-0.6), it occasionally assigns probabilities slightly above the market-implied probabilities. Nevertheless, across most events, it displays significant conservatism, and particularly at the higher end of the probability spectrum, Claude Sonnet 4 also is reluctant to place extreme probability predictions. Overall, while conservatism is a common trait across models, the extent of hesitation varies, with some models exhibiting considerably stronger reluctance than others. Similar trends across a wider range of models are reported in Appendix D.6.

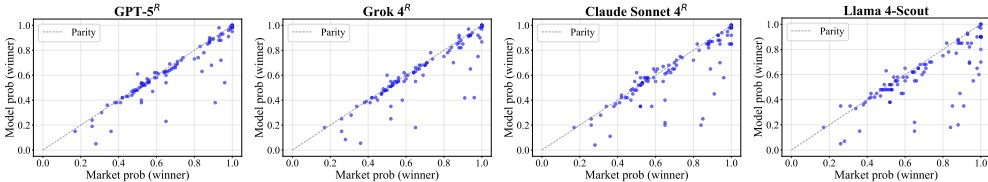

Figure 5: **LLM prediction probabilities compared to market baselines.** Scatter plots show the relationship between market-implied probabilities and predicted probabilities for various models. "Winner" refers to the market which resolves to `Yes`. Points on the diagonal line correspond to events where the model prediction matches the market-implied prediction probability for the `Yes` outcome markets.

### 4.1.4 WHICH CAPABILITIES ARE NEARLY MATURE?

In addition to the core experiments, we also examined two fundamental capabilities: (i) *robustness of probability elicitation*, where calibration remains stable under prompt variations and alternative probability estimation methods, and (ii) *logical consistency*, where most LLMs correctly understand structures such as mutually exclusive or nested markets. For the majority of models, both capabilities appear already reliable and largely mature. Detailed results are provided in Appendices D.3 and D.4.

### 4.2 LONGITUDINAL ANALYSIS OF LLM-AS-A-PROPHET

Forecasting is fundamentally temporal: we observe both the market and source dynamics changing as events approach their resolution time. `Prophet Arena` schedules predictions across multiple forecast horizons, allowing us to assess how predictive quality evolves with lead time (Fig. 6). As shown in Fig. 6, the average Brier score is plotted across lead-time intervals, where, for instance, the "0-3 h" bin represents predictions made within three hours before resolution. For both the market baseline and representative LLMs, accuracy generally improves as the resolution time approaches – reflecting that additional information becomes available and predictive signals strengthen.[7] Interestingly, the market baseline lags behind several frontier LLMs when predictions are made far in advance, suggesting that LLMs can effectively synthesize broader prior knowledge and reason under noisy settings at long horizons. However, as resolution nears, markets incorporate information and news updates more rapidly than LLMs, quickly surpassing LLMs in short-term accuracy.

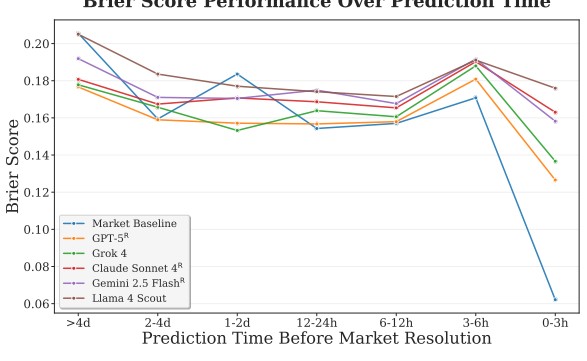

Figure 6: **Average Brier score across prediction lead times.** Scores are binned by time-to-resolution, with each point representing the mean Brier score of all forecasts falling within that window.

---

[7]In certain domains, such as live sports, markets may remain open during the event itself, allowing traders to incorporate real-time developments as outcomes unfold.

### 4.3 Granular Analysis of LLM-as-a-Prophet

The reported probabilities of LLMs compress a rich decision-making process into a single number: two models may yield similar predicted probabilities and scores while relying on dramatically different reasoning processes. To address this, we open the black box and evaluate the process underlying each forecast. Specifically, we adopt an LLM-as-a-judge framework (Zheng et al., 2023) to assess the soundness of reasoning along five critical dimensions: source selection, evidence extraction, reasoning synthesis, reasoning-to-prediction alignment, and recognition of prediction uncertainty. Full prompts and additional details of the evaluation framework are provided in Appendix D.7. To assess the reliability of the LLM-as-a-judge framework, we conducted a blinded human evaluation on a random subset of 170 predictions made by the LLMs. Human raters scored the same five dimensions using the identical rubric provided to the LLM judge. Overall, human and LLM ratings were closely aligned (mean difference $< 0.5$ out of a maximum of 4, with standard deviation $\approx 0.3$ across all dimensions). Full details of the study design and results are provided in Appendix D.7.1.

As shown in Table 3, the models demonstrate broadly comparable performance in source utilization, evidence extraction, and uncertainty analysis. However, substantial disparities emerge in the reasoning synthesis and reasoning-prediction alignment categories. For example, when comparing GPT-5 with Gemini 2.5 Flash, the differences in reasoning synthesis (0.95) and reasoning-to-prediction alignment (0.30) are markedly larger than the relatively minor gaps in source use (0.12), evidence extraction (0.00), and uncertainty analysis (0.20). Due to the significant deficiencies in those two key categories, the other models demonstrate a significant gap in prediction quality relative to GPT-5, supporting the findings in Table 2. These findings indicate a potential ceiling effect: once models attain proficiency in retrieval and evidence extraction, further performance gains depend primarily on advances in higher-order reasoning rather than incremental improvements in information access.

| LLM | Sources | Evidence | Reas. Synth. | Align. | Uncert. | Average Score |
|---|---|---|---|---|---|---|
| GPT-5[R] | **3.69** | **3.66** | **4.14** | **3.97** | **3.94** | **3.88** |
| Gemini 2.5 Flash[R] | 3.57 | **3.66** | 3.19 | 3.67 | 3.74 | 3.57 |
| Grok 4 | 3.40 | 3.51 | 3.33 | 3.48 | 3.66 | 3.48 |
| Claude Sonnet 4[R] | 3.53 | 3.47 | 2.93 | 3.39 | 3.75 | 3.41 |
| Llama 4 Scout | 2.97 | 2.88 | 2.29 | 2.37 | 2.87 | 2.68 |

Table 3: **LLM performance on reasoning evaluation criteria across dataset events.** Each dimension is scored on a standardized 5-point scale, where 1 and 5 indicate poor and excellent performance, respectively. Average scores are presented for each model, with **bold** values indicating the best-performing model for each criterion. Models are ordered by descending overall average score.

## 5 Conclusion

This paper systematically evaluates the prospects and challenges of using LLMs to forecast future events, a paradigm coined LLM-as-a-Prophet. Towards that end, we build `Prophet Arena`, a benchmark that allows modularized analysis about various aspects of existing LLMs' predictive intelligence. Our thorough experiments demonstrates the promise of LLM-as-a-Prophet reflected in the small forecasting loss, calibration error, strong reasoning synthesis and alignment of frontier models. However, we also identify key bottlenecks and highlight avenues for further progresses, such as better curation of context sources, more accurate internalization of knowledge and improving forecasts near events' resolution.

In addition, our findings raise deeper questions about the nature of predictability in forecasting tasks. Predictability depends not only on model capability, but also on the underlying information structure of events—what signals exist, how early they appear, how noisy they are, and how strongly they constrain outcomes. Our current metrics alone do not reveal whether performance gains stem from better signal extraction, stronger reasoning under uncertainty, or from tasks that are intrinsically more predictable. Interpreting forecasting results therefore requires disentangling task-level predictability from model-level capability.

Finally, we view Prophet Arena not merely as a leaderboard, but as a framework for probing predictive intelligence in context. The benchmark implicitly evaluates intertwined components: signal identification, evidence synthesis, uncertainty modeling, and probabilistic calibration. A key direction for future work is to more clearly separate these factors, clarifying what makes events predictable in principle and which model behaviors enable more accurate forecasts.

## ETHIC STATEMENT

Our evaluation pipeline of LLMs involve publicly available prediction market questions concerning widely known real-world events (e.g., sports, politics, entertainments). While we open-source a small subset of these events for transparency and reproducibility, no private, identifiable, or sensitive data is released and no human subjects were involved. The study adheres to the ICLR Code of Ethics by upholding standards of scientific integrity, transparency, and responsible research practice. We do not anticipate our methods or findings to pose risks of harm, privacy violations, or fairness concerns. While our findings may inform understanding of LLMs' probabilistic forecasting capabilities, we explicitly caution that they must not be construed as endorsements of specific applications nor as guarantees of real-world reliability.

## REPRODUCIBILITY STATEMENT

We have taken steps to ensure the reproducibility of our benchmark and experiment results. In Section 4, we provided the link to a curated subset of 100 evaluation events (PROPHET-ARENA-SUBSET-100) open-sourced on Huggingface. We also intend to expand this dataset in the future. All language models tested in our study are accessible through publicly available APIs and a complete list of the models and versions used are provided in Appendix B.6. The exact prompts used for all experiments are also included throughout Appendix D. Additional implementation and experimental details are described in the main text and appendices so as to facilitate independent verification and extensions of our work.

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

# ♣ APPENDIX: TABLE OF CONTENTS

## A    FULL DISCUSSION OF RELATED WORK

**Understanding and Advancing LLMs' Forecasting Capabilities.**   A key goal of our work is to understand the novel paradigm of *LLM-as-a-Prophet* and analyze how different capabilities (e.g., knowledge internalization, source usage, etc.) affect LLMs' predictive intelligence. This research goal shares similarities to a few recent works on understanding and diagnosing *special aspects* of LLMs' forecasting capabilities. For instance, Dai et al. (2025); Zhu et al. (2025) study LLMs' *temporal generalization* capability by challenging LLMs to forecast future events curated from news articles. Both works show that LLMs' forecasting accuracy degrades over time, even when they are armed with retrieval augmented generation (RAG). Paleka et al. (2025b) studies whether LLMs can make consistent forecasts; for example, a logical AI should not predict that both the Democratic and Republican parties have a 60% chance of winning the 2024 US presidential election. Towards that end, they build a proper-scoring-rule forecasting benchmark to measure the consistency of LLMs' predictions. Similar to both works' research methodology, we also build a benchmark platform `Prophet Arena` to study our research question. However, to our knowledge, our work is the first to investigate the general predictive intelligence of the LLM-as-a-Prophet paradigm. Finally, we also note the recent line of works advancing the forecasting capabilities of AI systems using language models (Zou et al., 2022; Halawi et al., 2024). Though our goal of benchmarking is different, we envision that insights from our analysis about the strengths and limitations of current LLMs for forecasting could benefit future research for advancing AI's forecasting capabilities.

**Forecasting Benchmarks.**   Forecasting has recently become a popular challenge for benchmarking LLMs' capabilities. Besides being a real challenge to LLMs, it also avoids the thorny issue of benchmark contamination due to the evaluating LLMs on "future" events (Dai et al., 2025; Karger et al., 2025). To our knowledge, (Jin et al., 2021) is perhaps one of the earliest to test such forecasting capabilities of language models. They introduced ForecastQA, an evaluation dataset consisting of 10,392 crowdsourced multiple-choice questions, and the performance of language models at the time (mainly BERT models) still significantly lags behind human performance. Since Jin et al. (2021), there has been a progressive line of work developing more and more challenging forecasting benchmarks for more advanced models, by integrating prediction market events (Zou et al., 2022), extracting events from news articles (Zhang et al., 2024; Wang et al., 2025), curating future-oriented questions from websites (Wildman et al., 2025), using open-ended queries (Guan et al., 2024), and lately developing dynamic benchmarks with live event and leaderboard updates (Zeng et al., 2025; Karger et al., 2025; Bianchi et al., 2025).

We also build a benchmark `Prophet Arena`, but as mentioned above, the main goal of our work is to understand LLM-as-a-prophet and analyze how its different capabilities affect the forecasting. We built `Prophet Arena` primarily for this analysis purpose, similar to that of (Dai et al., 2025; Paleka et al., 2025b). Towards this end, our work departs from these benchmark researches in multiple ways. First, we focus on comprehensive evaluation about LLMs' predictive intelligence from various dimensions such as the Brier score, calibration and economic values of forecasts. In Section 3.1, we illustrate how these metrics differ and when each metric should be preferred. In contrast, most previously works mainly focus on one metric, such as Brier score Halawi et al. (2024); Karger et al. (2025), calibration Zou et al. (2022) and accuracy Zeng et al. (2025); Dai et al. (2025), rendering them not full capture the quality of forecasting (as we also illustrate in Section 3). Second, because our goal is not to rank LLMs or agents as in the above benchmark papers but rather to understand frontier LLMs' forecasting capabilities, we need a natural baseline forecaster to know how good frontier LLMs are. Towards that end, we build `Prophet Arena` based on events from prediction markets, and hence can use the market's forecasts as a cost-effective and justified baseline forecaster (Arrow et al., 2008). Finally, besides reporting different evaluation metrics, we further dive into the LLM-as-a-Prophet paradigm and offer insights about how the forecasts are affected by different capabilities such as knowledge internalization and source usage.

## B    DEFINITIONS AND PROPHET ARENA PIPELINE DETAILS

### B.1    OVERVIEW

We summarize the core components of the `Prophet Arena` pipeline, which together distinguish it from prior forecasting benchmarks (see Table 4 for a comparison with existing approaches).

**(1)   Event and Market Extraction.**   `Prophet Arena` collects unresolved events from Kalshi, a live prediction platform with events spanning across categories such as finance, sports, politics

and entertainment. Kalshi provides structured, real-time events with verifiable outcomes, enabling standardized and comparable evaluation across models. To ensure informativeness and comparability, we filter by *Popularity* (volume/liquidity/volatility), *Diversity* (domain balance), and *Recurrence* (repeated formats). `Prophet Arena` periodically retrieves a number of unresolved events, making it a truly live benchmark, rather than a fixed, static dataset susceptible to training data contamination. The use of prediction markets guarantees that all questions pertain to genuine future outcomes grounded on real world importance and that their resolutions can be objectively verified once finalized.

**(2) Prediction Context Construction.**

For each event, `Prophet Arena` constructs a unified *prediction context* that all models receive identically. This context contains: (1) Relevant information sources retrieved by an LLM-based search agent using web queries that collect titles, snippets, timestamps, and URLs of recent news or reports; and (2) Market snapshots including the latest `Yes/No` contract prices and trading volumes, from which implied probabilities (based on market contract prices) are derived. Providing identical contexts isolates differences in models reasoning and calibration rather than their retrieval capabilities. The search component in `Prophet Arena` is fully *searcher-agnostic*: new search agents can be added or replaced without altering the forecasting protocol or evaluation procedure. In all experiments presented in this paper, we use a single LLM-based searcher instantiated with GPT-4o equipped with web access.

**(3) Probabilistic Forecasting and Comprehensive Evaluation.** Given an event and its constructed context, each model produces a probabilistic forecast for every market within the event, outputting a predicted probability for a `Yes` resolution alongside a natural-language rationale. This process is highly *modularized*, decomposing prediction into distinct stagesfrom source collection and context construction to probability elicitationenabling controlled experimentation and analysis of performance at each module. To capture the temporal evolution of forecasts, we implement a *multi-horizon protocol* where models predict at various timestamps as market conditions and public information evolve. While all rationales are logged for qualitative analysis, only the probabilities are used for quantitative evaluation once the event resolves. By evaluating these forecasts against *multiple complementary metrics* detailed in Section 3, `Prophet Arena` transforms live, real-world questions into a continuous, contamination-resistant benchmark for predictive intelligence.

**(*) Bonus feature: Market Baseline.** To establish a fair and interpretable anchor for comparing different LLM forecasters, we include a *Market Baseline* – a synthetic forecaster whose prediction is defined as the market-consensus probability that market will resolve to `Yes`. In practice, this probability is inferred from the normalized contract prices. [8] As we will demonstrate in Section 3, this market baseline serves as an informative benchmark for assessing *forecast difficulty* and contextualizing model performance. When LLMs outperform the market baseline, they demonstrate genuine predictive advantage over the aggregated human consensus in the real-world market.

| Benchmark | Live events | Probabilistic | Multi-horizon | Modularized | Robust metrics |
|---|---|---|---|---|---|
| MIRAI | – | – | ✓ | – | – |
| FORECASTBENCH | ✓ | ✓ | ✓ | – | – |
| FUTUREBENCH | ✓ | – | – | – | – |
| FUTUREX | ✓ | – | – | – | – |
| **Prophet Arena** | ✓ | ✓ | ✓ | ✓ | ✓ |

Table 4: **Comparisons with related forecasting benchmarks**, including MIRAI (Ye et al., 2024), ForecastBench (Karger et al., 2025), FutureBench (Bianchi et al., 2025), FutureX (Zeng et al., 2025).

### B.2 EVENT EXTRACTION

`Prophet Arena` sources unresolved events from Kalshi, a live prediction platform with events spanning across finance, sports, politics, sports, and entertainment. To ensure informativeness and comparability, we filter by *Popularity* (volume/liquidity/volatility), *Diversity* (domain balance), and *Recurrence* (repeated formats). `Prophet Arena` periodically retrieves 20 unresolved events each day at 12 AM (UTC).

---

[8]In practice, contract prices may not sum up to one due to exchange fees, requiring slight normalization.

**Event.** Defined formally, let $\{E_i\}_{i \in [K]}$ denote the set of evaluated *events*. An *event* is the overarching question or subject concerning a future real-world occurrence. It serves as a high-level container for one or more tradable *markets*. In many prediction markets, an event itself is **not** a tradable asset; rather, it sets the context, scope, and resolution criteria for the markets that fall under it.

- **Example 1:** "Who will win the 2025-26 NBA Championship?"
- **Example 2:** "Which individuals will President Trump officially meet in 2025?"

**Market.** Each event $E_i$ contains *markets* $\{M_{ij}\}_{j \in [N_i]}$. A *market* is a specific, tradable proposition under an event that resolves to a definitive `Yes` or `No` outcome. Each market represents a potential, verifiable answer to the event's overarching question. For a given market, a *`Yes` contract* is a 0-1 random variable that achieves value 1 if the `Yes` outcome is realized, and 0 otherwise. A *"NO" contract* is defined similarly, and always pays out in the opposite direction as the "YES" contract.

- **Under Event 1, a market could be:**
  "The Boston Celtics will win the 2025 NBA Championship."

- **Under Event 2, a market could be:**
  "President Trump will officially meet with Emmanuel Macron in 2025."

**Event Resolution.** An event $E_i$ resolves at time $\tau_i$ with realized outcome indicator $o_{ij} \in \{0, 1\}$, where $o_{ij} = 1$ means the corresponding market $M_{ij}$ of the event is realized (`Yes`). When the event index $i$ is clear from context, we often drop the subscript $i$ and write $M_j, o_j$.

### B.3 PREDICTION CONTEXT

**Prediction scheduling.** For each event $E_i$, the benchmark specifies a finite set of pre-resolution forecast times (horizons) $\mathcal{T}_i \subset (-\infty, \tau_i)$. We construct $\mathcal{T}_i$ by placing each subsequent forecast halfway between the current forecast time and the event close time $\tau_i$. Let the first forecast time be $t_i^{(0)} < \tau_i$ and define the initial gap $\Delta_i^{(0)} := \tau_i - t_i^{(0)} > 0$. For $k \geq 0$,

$$t_i^{(k+1)} = \frac{t_i^{(k)} + \tau_i}{2} \quad \Longleftrightarrow \quad t_i^{(k)} = \tau_i - 2^{-k}\,\Delta_i^{(0)}, \quad \Delta_i^{(k+1)} = \frac{\Delta_i^{(k)}}{2}.$$

To avoid excessive clustering near $\tau_i$, we enforce a minimum time gap $\delta_{\min} > 0$ between the last forecast and the close time.

**Context construction.** For each event $E_i$ and forecasting time $t \in \mathcal{T}_i$ we construct a curated context $C_{i,t}$ shared across models; this isolates forecasting ability in $p_{ij,t}$ from retrieval variability. The context $C_{i,t}$ consists of two components: relevant news sources and market data. The relevant news sources are retrieved by LLM searchers. The prompt for search is shown below. The market snapshot is fetched from Kalshi API, and contains three fields for each market: `last_price` (price of the last transaction), `yes_ask` (asking price for buying `Yes`), `no_ask` (asking price for buying `No`). From that snapshot we extract the *implied probability* $q_{ij,t} \in [0, 1]$ for a `Yes` contract at time $t$ (with `No` priced at $1 - q_{ij,t}$).[9]

At each forecasting time $t_i \in \mathcal{T}_i$, LLM searchers will be dispatched and live market snapshot will be retrieved. Together, relevant news sources from LLM searchers and market snapshots from Kalshi will serve as the prediction context.

Importantly, `Prophet Arena` is *searcher-agnostic*: the search component is an independent, pluggable module and adding new searchers does not change the forecasting protocol or the scoring of $p_{ij,t}$. `Prophet Arena` actively updates to include new LLM searchers. In this paper, we instantiate a single LLM searcher using GPT-4o with web search enabled and all experiments use that configuration (Appendix F.1.1).

### B.4 PROBABILISTIC FORECASTING TO ACCOUNT FOR UNCERTAINTY

**LLM-predicted Probabilities.** Given $(E_i, M_{ij}, C_{i,t}, t)$, a model must output an *LLM probability*,

$$p_{ij,t} \in [0, 1],$$

---

[9]Implied probabilities are reverse-engineered from contract prices; transaction fees may cause `YES`/`NO` prices not to sum to 1.

interpreted as its belief that $M_{ij}$ will realize Yes at time $t$, accompanied by a natural-language rationale (logged for analysis but not used in scoring). The prediction prompts are documented in Appendix F.1.2.

**Implied Probabilities.** At each $t \in \mathcal{T}_i$, a Yes (resp. No) contract is valued at $q_{ij,t}$ (resp. $1 - q_{ij,t}$). The *implied probability* $q_{ij}$ represents the (human) market-consensus belief that the Yes outcome will come true.

## B.5   RESOLUTION AND EVALUATION

**Edge and Utility.** We denote a (yes) *edge* $e_{ij} := \frac{p_{ij}}{q_{ij}}$ as the likelihood-ratio between the LLM-predicted and implied probabilities. A larger edge signals the LLM to be more confident (than the market) that a market will be realized. Similarly, we define the (no) edge to be $\tilde{e}_{ij} := \frac{1-p_{ij}}{1-q_{ij}}$.

We assume that the *price* of a single Yes contract simply equals the implied probability $q_{ij}$, and the price of a single No contract is thus $1 - q_{ij}$.[10] In the sequel, our strategy is limited to buying contracts (taking a long position). But all contracts can be purchased in fractional amounts.

Our simulated trading policies (used only for *relative* metrics) considers the utility function with risk-aversion hyperparameter $\gamma \in [0, 1]$. It maps any wealth (i.e. payoff) to a utility.

**Prediction Evaluation.** After the close time $\tau_i$, we will retrieve the outcomes of event $E_i$ from Kalshi. For each market $M_{ij}$ and each horizon $t \in \mathcal{T}_i$, we score $(p_{ij,t}, o_{ij})$ with *absolute* proper scoring rules (e.g., Brier-based) and, separately, evaluate *relative* Average Return by simulating trades against $q_{ij,t}$ under $U_\gamma$. Scores are then aggregated across $j$, $i$, with the specific calculations detailed in the subsequent evaluation section.

The below table summarizes the notations that will appear in the later math expressions:

| | |
|---|---|
| $E_i$, | $i \in [K]$, the $i$-th event in our evaluation set. |
| $M_{ij}$ $(M_j)$ | $j \in [N_i]$, the $j$-th market of the $i$-th event, we drop the $i$ subscript when the market is obvious (same for below). |
| $p_{ij}$ $(p_j)$ | the LLM-predicted probability that $M_{ij}$ will realize. |
| $q_{ij}$ $(q_j)$ | the market implied probability that $M_{ij}$ will realize. |
| $e_{ij}/\tilde{e}_{ij}$ $(e_j/\tilde{e}_j)$ | the (yes/no) edge of $M_{ij}$. |
| $o_{ij}$ $(o_j)$ | the indicator of whether $M_{ij}$ is realized. |
| $U_\gamma(w)$ | the utility function with risk-aversion hyperparameter $\gamma \in [0, 1]$. It maps any wealth (i.e. payoff) to a utility. |

## B.6   COMPREHENSIVE LIST OF EVALUATED LLMS

**Remark.** The column **Default Reasoning** specifies the reasoning configuration used when a model is referenced without qualifiers (e.g. GPT-5, or when we write GPT-5$^\text{R}$ in Table 2). In the full evaluation table (Table 7), some models appear multiple times under different reasoning settings; each such variant is treated as a distinct model (e.g., GPT-5 (Minimal)).

Reasoning configuration is inherently model-dependent. For "hybrid reasoning" models that allow toggling between thinking and non-thinking modes, the configuration is specified as either *enabled* or *disabled*. For models that expose explicit control over reasoning effort, the configuration is expressed in levels (e.g., *minimal*, *medium*, *high*).

---

[10]In practice, (1) we actually "reverse-engineer" the implied probabilities from market contract prices, and (2) the prices of Yes/No contracts might not sum to 1 due to transaction fees taken by the exchange.

| LLM | Citation | Open Weight? | Default Reasoning |
|---|---|---|---|
| GPT-5 | OpenAI (2025b) | No | High |
| o3 | OpenAI (2025c) | No | High |
| o3-Mini | OpenAI (2025c) | No | Medium |
| o4-Mini | OpenAI (2025c) | No | High |
| Gemini 2.5 Pro | Comanici et al. (2025) | No | Enabled |
| Gemini 2.5 Flash | Comanici et al. (2025) | No | Enabled |
| Grok-4 | xAI (2025b) | No | Enabled |
| Grok-3-Mini | xAI (2025a) | No | Enabled |
| GPT-4.1 | OpenAI (2025a) | No | N/A |
| Claude Sonnet 4 | Anthropic (2025) | No | Enabled |
| Kimi-K2 | Kimi Team (2025) | Yes | N/A |
| GPT-4o | OpenAI (2024) | No | N/A |
| Llama 4 Maverick | Meta Team (2025) | Yes | N/A |
| Llama 4 Scout | Meta Team (2025) | Yes | N/A |
| DeepSeek-V3 | Liu et al. (2024) | Yes | N/A |
| DeepSeek-R1 | Guo et al. (2025) | Yes | Enabled |
| Qwen3-235B | Yang et al. (2025) | Yes | N/A |
| Gemini 2.0 Flash | Gemini Team (2024) | No | N/A |
| Gemini 2.0 Flash (Lite) | Gemini Team (2024) | No | N/A |

Table 5: **Comprehensive list of LLMs evaluated in** `Prophet Arena` (as of submission time).

## C TECHNICAL DETAILS

### C.1 FURTHER INTUITIONS BEHIND BRIER SCORE

In `Prophet Arena`, the set of markets $\{M_{ij}\}_{j=1}^{m_i}$ under an event $E_i$ need not be mutually exclusive. This has two implications: (1) the realized outcome vector $\mathbf{o}_i = (o_{i1}, \ldots, o_{im_i})'$ may contain multiple ones rather than being strictly one-hot, and (2) the predicted probabilities $\{p_{ij}\}_j$ may sum to more than one. Our Brier score formulation remains robust to these cases because the score is always evaluated at the *market level*, where each individual market is binary – resolving either to `Yes` or `No`, but not both. If we pool all markets across all events and relabel them $M_1, \ldots, M_k$, the standard binary Brier score is simply

$$BS = \frac{1}{k} \sum_{j=1}^{k} (p_j - o_j)^2. \tag{1}$$

Our event-level definition in Section 3 can be viewed as a *weighted* version of Eq. (1). Since events vary greatly in the number of associated markets, directly pooling them would let large events dominate the metric. To mitigate this, we assign each market a weight $w_{ij} = 1/m_i$, inversely proportional to the number of markets in its event. This yields the final form:

$$BS = \frac{1}{\sum_i \sum_j w_{ij}} \sum_{i=1}^{N} \sum_{j=1}^{m_i} w_{ij}(p_{ij} - o_{ij})^2 = \frac{1}{N} \sum_{i=1}^{N} \left( \frac{1}{m_i} \sum_{j=1}^{m_i} (p_{ij} - o_{ij})^2 \right).$$

This weighting ensures that each event contributes equally, regardless of its size, while still respecting the binary resolution of individual markets.

### C.2 EXPECTED CALIBRATION ERROR(ECE) AND ITS EMPIRICAL ESTIMATION

In our binary market setting, let $M = \{1, \cdots, m\}$ denote the set of markets. Suppose a forecaster provides predicted probabilities $\tilde{p}_k$ for all markets in a set $M_k \subset M$ ($m_k = |M_k|$ denotes its cardinality). Then the **expected calibration error (ECE)** of this forecaster is formally defined as

$$ECE = \frac{1}{m} \sum_k \left| \sum_{j \in M_k} \mathbb{P}(o_j = 1 | p_j = \tilde{p}_k) - m_i \tilde{p}_k \right|. \tag{2}$$

Intuitively, ECE captures how much a probabilistic forecast differs from the real averaged probability, given the forecast. This formal definition of ECE is in terms of true conditional probabilities. In practice, this definition cannot be computed exactly, as the conditional terms $\mathbb{P}(o_j = 1 \mid p_j = \tilde{p}_i)$ are not directly observable. Instead, the standard approach in the applied literature is to approximate ECE via a binned empirical estimate.

Formally, let $\{(p_j, o_j)\}_{j=1}^m$ denote a set of predicted probabilities and realized outcomes. We first partition the unit interval $[0, 1]$ into $B$ disjoint bins $\{I_b\}_{b=1}^B$, and assign each prediction $p_j$ to its corresponding bin. Let $M_b = \{j : p_j \in I_b\}$ be the index set of predictions falling into bin $b$, and $m_b = |M_b|$ its size. Define

$$\hat{p}_b = \frac{1}{m_b} \sum_{j \in M_b} p_j, \quad \hat{o}_b = \frac{1}{m_b} \sum_{j \in M_b} o_j,$$

as the average predicted probability and empirical frequency of `Yes` outcomes in bin $b$, respectively. The **empirical expected calibration error** is then given by

$$\widehat{ECE} = \frac{1}{m} \sum_{b=1}^B m_b \cdot \left| \hat{o}_b - \hat{p}_b \right|. \tag{3}$$

Intuitively, $\widehat{ECE}$ measures the weighted average discrepancy between empirical accuracy and average predicted probability across bins, with weights proportional to bin counts. Throughout our experiments, all reported calibration results correspond to this empirical version (with $B = 10$).

### C.3 A Unified Framework for Utility-Maximizing Betting Strategy

In Section 3, we introduced **Average Return** which captures expected reward under a risk-neural investor's optimal betting strategy. Here we offer more details about how such a betting strategy can be characterized. Formally, consider a (binary) market with market price $q_Y$ per share for the `Yes` contract and $q_N$ for the `No` contract. Given forecasted probability $p$ for an `Yes` outcome, the following natural betting strategy turns out to be optimal for a risk-neutral investor: allocate a unit budget (\$1) to buy $1/q_Y$ shares of `Yes` contracts if $\frac{p}{q_Y} \geq \frac{1-p}{q_N}$, or to buy $1/q_N$ shares of `No` contracts otherwise. After the event resolution, if the bought contracts match the event outcome, the return is simply the number of these contracts; otherwise, the return is $0$.

To more systematically understand the optimal betting strategy, next we develop a unified framework for betting strategies in binary markets for risk-sensitive investors. This framework serves two purposes:

1. It formalizes betting as a utility-maximization problem, allowing us to flexibly encode different risk preferences.
2. It shows that the simple strategy mentioned above is a special case of this unified framework, corresponding to the risk-neutral setting.

**Contracts and market prices.** Each binary market resolves to either `Yes` or `No`. A share of a `Yes` contract pays \$1 if the outcome is `Yes` and \$0 otherwise; the price of this contract is denoted $q$. Symmetrically, a share of a `No` contract costs $1 - q$ and pays \$1 if the outcome is `No`. These prices are often interpreted as market-implied probabilities (Wolfers & Zitzewitz, 2006), though our framework does not rely on this interpretation.

**Utility function.** Let $U : \mathbb{R}_{\geq 0} \to \mathbb{R}$ denote a utility function mapping a payoff $w$ to its perceived value (satisfaction) $U(w)$. We focus on the class of constant relative risk aversion (CRRA) utilities:

$$U_\gamma(w) = \begin{cases} \dfrac{w^{1-\gamma}}{1-\gamma}, & 0 \leq \gamma < 1, \\ \log w, & \gamma = 1 \end{cases} \tag{4}$$

where $\gamma \in [0, 1]$ indexes risk aversion. At $\gamma = 0$, $U_\gamma$ is linear (risk-neutral); at $\gamma = 1$, it reduces to $\log w$ (log utility, risk-averse); and for $\gamma \in (0, 1)$, it interpolates smoothly between the two. Fig. 7 visualizes representative cases.

**Budget allocation as optimization.** Fix a unit budget of \$1 for each market. Let $a_Y$ and $a_N$ denote the amounts allocated to `Yes` and `No` contracts, with $a_Y + a_N = 1$. Purchasing $a_Y$ dollars of `Yes`

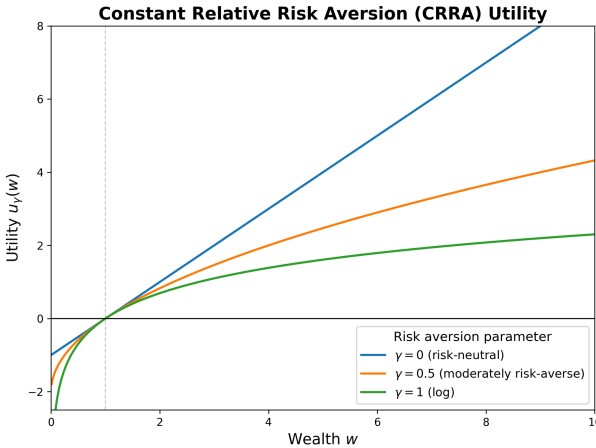

Figure 7: **Visualization of CRRA utility functions.**

contracts yields $a_Y/q$ shares (i.e., the pay off is $a_Y/q$ if the outcome is Yes); similarly, $a_N/(1-q)$ shares of No pay off if the outcome is No. Given belief (predicted probability) $p$ and risk preference $\gamma$, the optimal allocation is the solution to

$$\max_{a_Y, a_N \geq 0 \,:\, a_Y + a_N = 1} \left[ p \cdot U_\gamma \left( \frac{a_Y}{q} \right) + (1-p) \cdot U_\gamma \left( \frac{a_N}{1-q} \right) \right]. \tag{5}$$

This objective is the expected utility of betting under $p$.

**Closed-form solutions.** For $\gamma > 0$, the optimization admits a closed-form solution:

$$a_Y^* \equiv a_Y^*(\gamma) = \frac{q^{1-\frac{1}{\gamma}} \, p^{\frac{1}{\gamma}}}{q^{1-\frac{1}{\gamma}} \, p^{\frac{1}{\gamma}} + (1-q)^{1-\frac{1}{\gamma}} \, (1-p)^{\frac{1}{\gamma}}}, \qquad a_N^* \equiv a_N^*(\gamma) = 1 - a_Y^*, \tag{6}$$

At $\gamma = 0$, we take the limit $\gamma \to 0^+$, yielding

$$a_Y^*(0) \equiv \lim_{\gamma \to 0} a_Y^*(\gamma) = \begin{cases} 1, & \text{if } p > q, \\ 0, & \text{if } p \leq q \end{cases}.$$

which recovers the strategy described at the beginning of this subsection. At the other extreme, when $\gamma = 1$, the optimal strategy is to allocate $a_Y^* = p$ and $a_N^* = 1-p$, i.e., to bet proportionally to ones own probabilities, independent of the market price. Overall, this unified framework highlights how different betting strategies arise from different risk preferences. The "all-in" rule in the main text is simply the risk-neutral optimum.

### C.4 BRIER SCORE VS ECE.

As discussed in Section 3, calibration errors and Brier scores measure different aspects of forecasts. We start with a simple example to illustrate that a better-calibrated forecast can have a worse Brier score. Suppose there are two markets $M_1, M_2$ with ground-truth probability $p_1^* = 0.9, p_2^* = 0.1$. Alice's prediction is $p_1^A = 1, p_2^A = 0$ whereas Bob's prediction is $p^B = 0.5$ for both markets. It is not difficult to see that Alice has a better/smaller Brier score, but she turns out to have a worse/larger ECE. Specifically, since Alice has two different probability predictions, her ECE is $\frac{1}{2}(|p_1^* - p_1^A| + |p_2^* - p_2^A|) = 0.1$ whereas Bob has a single probability prediction with the ECE equaling $\frac{1}{2}|p_1^* + p_2^* - 2p^B|| = 0$. That is, Bob's prediction of $0.5$ probability is perfectly calibrated and reliable as, conditioned on this probability, events' average probability is indeed $0.5$.

**A forecaster with smaller ECE but larger Brier score induces better risk-adjusted decision making.** Consider an event $E$ with binary outcome $o \in \{0, 1\}$ and a ground-truth probability $p^* = 0.5$ to have $o = 1$. Suppose forecaster $A$ predicts probability $p_A := \Pr(o = 1)$ which

correlates with the realized outcome $o$ in the following way

$$(p_A, o)) = \begin{cases} (1,1), & \text{with joint probability } 0.5(1-\epsilon) \\ (0,0), & \text{with joint probability } 0.5(1-\epsilon) \\ (1,0), & \text{with joint probability } 0.5\epsilon \\ (0,1), & \text{with joint probability } 0.5\epsilon \end{cases}$$

where $\epsilon$ is a small value. In other words, forecaster $A$ predicts $p_A = 1$ or $p_A = 0$, each with probability 0.5 and with $\epsilon$ fraction of errors. These are extremal and uncalibrated forecasts but is almost always correct. It is easy to verify that the Brier score of $p_A$ is $0.5\epsilon \times (1-0)^2 + 0.5\epsilon \times (1-0)^2 = \epsilon$, whereas expected calibration error (ECE) is $0.5 \times [1-(1-\epsilon)]^2 + 0.5 \times [1-(1-\epsilon)]^2 = 0.5\epsilon^2$.

Next consider forecaster $B$ predicts probability $p_B := \Pr(o = 1)$ which correlates with the realized outcome $o$ in the following way

$$(p_B, o)) = \begin{cases} (2/3, 1), & \text{with joint probability } 0.5 \\ (2/3, 0), & \text{with joint probability } 0.25 \\ (0, 0), & \text{with joint probability } 0.25 \end{cases}$$

That is, forecaster $B$ predicts $p_B = 2/3$ for 75% of the time, and predicts $p_B = 0$ otherwise. It is easy to see that $p_B$ is perfectly calibrated with $ECE = 0$, but has Brier score equal $0.75 \times [\frac{2}{3}(2/3 - 1)^2 + \frac{1}{3}(2/3 - 0)^2] = 1/6$.

Clearly, $p_B$ is more well-calibrated than $p_A$ but has much worse Brier score, especially when $\epsilon$ is small. However, we show that $p_B$ is better for risk-averse decision making, than $p_A$. Suppose a decision maker has 1 unit of fund and would like to buy contracts for event $E$'s outcome. Suppose the current market price for the contract $o = 1$ and $o = 0$ are both 0.5 per contract. Adopting a classic risk-averse utility function, we assume her utility $U(x) = \log(x)$ (i.e., $x$ amount of return means $U(x)$ utility to her).

If the decision maker adopts forecaster $A$, since $A$ makes extremal forecasts, even under risk-averse utility $\log(x)$ her strategy is to spend all the 1 unit of fund to buy two units of contract $o = 1$ whenever $p_A = 1$, and buy 2 units of $o = 0$ otherwise. With $\epsilon$ probability of losing all her fund, her expected *risk-adjusted* utility is thus $(1-\epsilon)U(2) + \epsilon U(0)$, which goes to $-\infty$ when $U(x) = 0$, regardless how small $\epsilon$ is.

If the decision maker adopts the perfectly calibrated forecaster $B$, she would spend all fund to buy $o = 0$ contract when $p_B = 0$, but split her 1 unit of fund to buy $2B_1$ of $o = 1$ contracts and $2B_0$ of $o = 0$ contracts, so as to maximize her risk-adjusted utility as $\max_{B_0 + B_1 = 1} 2/3U(2B_1) + 1/3U(2B_0)$. Simple calculation shows that optimal $B_1 = 2/3$, leading to optimal risk-adjust utility $2/3U(4/3) + 1/3U(2/3) = \log(\frac{32}{27})/3$, which is strictly larger than her initial utility $\log 1 = 0$.

**Remarks.** A few remarks are worthwhile to mention about this example. First, forecaster $p_A$ with a small $\epsilon$ will be better for a risk-neural decision maker. However, in this case, while the decision maker will have larger expected utility, she will have to bear the possibility of losing all her 1 unit of fund, though also enjoy the possibility of doubling her fund to 2 units. This is often not desirable in real-world applications, such as financial decisions in which risk control is extremely important. In such cases, more well-calibrated forecaters are preferred. Second, our example has extremely negative utility $-\infty = U(0)$. This helps us to cleanly highlight the underlying conceptual message, but simple tweak to the utility function can remove that extreme situation yet arrive at the same conclusion.

### C.5 BRIER SCORES VS. MARKET RETURNS

We provide a simple example to prove the following fact.

*Fact* 1. *There exist a binary prediction market with two forecaster $A, B$ such that $A$ has strictly higher market return than $B$ but has strictly worse/higher Brier score.*

*Proof.* Suppose we bet on a single event with binary outcomes, with ground-truth probability of Yes being 0.6 (for the event to be realized), and prediction market price 0.5. Consider two different probabilistic predictions of this events Yes realization: model A predicts 0.45 and model B predicts 0.9.

The expected Brier Score of A is:

$$1 - \left[0.6 \cdot (0.45 - 1)^2 + 0.4 \cdot (0.45 - 0)^2\right] = 1 - 0.2625 = 0.7375$$

The expected Brier Score of B is:

$$1 - \left[0.6 \cdot (0.9 - 1)^2 + 0.4 \cdot (0.9 - 0)^2\right] = 1 - 0.33 = 0.67$$

So A has a higher Brier Score. However, because A predicts $0.45$, lower than the $0.5$ prediction market price, A will short `Yes` (or equivalently, buy `No`) at $0.5$. Meanwhile, B predicts $0.9$, much higher than the prediction market price, so B will buy `Yes`. Respectively, A and Bs expected returns will be:

$$0.6 \cdot (-1 + 0.5) + 0.4 \cdot (0.5) = -0.1$$
$$0.6 \cdot (1 - 0.5) + 0.4 \cdot (-0.5) = 0.1$$

Therefore, A has a higher Brier score, but lower returns.

$\square$

This example uncovers a key difference between the two metrics. The Brier Score measures how close a prediction is to the ground truth and, importantly, has nothing to do with market prices. Since As prediction above is closer to the ground truth, it receives a higher Brier Score. However, returns on the market are not only driven by the true probability but also by the market price. Therefore, even though Bs prediction is exaggerated, it lies on the correct side of the market mispricing (buying "Yes" when the outcome is more likely than price suggests), thereby achieving higher returns.

### C.6 MARKET RETURNS VS. CALIBRATION

In Section 3, we introduced the evaluation pipeline in `Prophet Arena`, including how market return is calculated based on market decisions (buying either the YES or NO contract), which are derived from predicted and market-implied probabilities $(p_j, q_j)$ as well as the realized outcome $o_j$ under the following decision rule: *buy YES contract whenever $p_j > q_j$, and buy NO otherwise.*

Here we further make the connection between a (perfectly) calibrated predictor and **market returns conditioning on the type of decision made**. Before stating the main result, the following mild assumptions and definitions are needed.

**Assumption C.1** (Data Generation). We assume that the triples $(o_j, p_j, q_j)$ are drawn i.i.d. from some joint probability distribution $\mathcal{D}$. The expectation $\mathbb{E}[\cdot]$ below is taken over $\mathcal{D}$.

**Assumption C.2** (Sufficiency of Calibrated Predictors). Perfect calibration implies that $\forall c \in [0, 1]$,

$$\mathbb{E}[o_j \mid p_j = c] = 1 \cdot \mathbb{P}[o_j = 1 \mid p_j = c] + 0 \cdot \mathbb{P}[o_j = 0 \mid p_j = c] = c.$$

We futher assume that a calibrated predictor is **sufficient** for the market outcome: the outcome $o_j$ is conditionally independent of the market price $q_j$ given the forecast $p_j$, i.e. $\mathbb{E}[o_j \mid p_j, q_j] \equiv \mathbb{E}[o_j \mid p_j]$

**Definition C.3** (Symmetric Disagreement). We say a predictor is **symmetric** (against the market) if the probability distribution of $d_j := p_j - q_j$ is **symmetric about zero**. We let $f(x)$ denote the p.d.f. of $d_j$ and the definition implies (1) $f(x) = f(-x)$ for all $x \in \mathbb{R}$, and (2) $\mathbb{P}[p_j > q_j] = \mathbb{P}[p_j \leq q_j]$. In other words, $d_j$ represents the "disagreement" that the predictor perceives over the market, and a **symmetric** predictor is not consistently more conservative or more aggressive.

**Theorem C.4** (Symmetric Expected Returns for Calibrated Predictors). *Let $R_Y = o_j - q_j$ be the return on a YES contract and $R_N = (1 - o_j) - (1 - q_j) = q_j - o_j$ be the return on a NO contract. For a predictor that is both **perfectly calibrated** and **symmetric** against the market, the expected returns conditioned on the betting decision are equal, i.e.*

$$\mathbb{E}[R_Y \mid p_j > q_j] = \mathbb{E}[R_N \mid p_j < q_j]. \tag{7}$$

*Proof.* Let $E_Y = \mathbb{E}[o_j - q_j \mid p_j > q_j]$ and $E_N = \mathbb{E}[q_j - o_j \mid p_j < q_j]$ denote the average returns on YES and NO contracts, respectively. We begin by analyzing the expected outcome $o_j$ under each condition. By the Law of Iterated Expectations and the sufficiency of the calibrated predictor (Assumption C.2), we can replace the outcome $o_j$ with the prediction $p_j$ inside the expectation:

$$\mathbb{E}[o_j \mid p_j > q_j] = \mathbb{E}\left[\mathbb{E}[o_j \mid p_j, q_j] \mid p_j > q_j\right] = \mathbb{E}[p_j \mid p_j > q_j],$$
$$\mathbb{E}[o_j \mid p_j < q_j] = \mathbb{E}\left[\mathbb{E}[o_j \mid p_j, q_j] \mid p_j < q_j\right] = \mathbb{E}[p_j \mid p_j < q_j].$$

Substituting these back into the expressions for $E_Y$ and $E_N$:

$$E_Y = \mathbb{E}[p_j \mid p_j > q_j] - \mathbb{E}[q_j \mid p_j > q_j] = \mathbb{E}[p_j - q_j \mid p_j > q_j],$$
$$E_N = \mathbb{E}[q_j \mid p_j < q_j] - \mathbb{E}[p_j \mid p_j < q_j] = \mathbb{E}[q_j - p_j \mid p_j < q_j].$$

The theorem is proven if we can show that $\mathbb{E}[p_j - q_j \mid p_j > q_j] = \mathbb{E}[q_j - p_j \mid p_j < q_j]$. Let $d_j = p_j - q_j$ be the disagreement variable. The equality becomes:

$$\mathbb{E}[d_j \mid d_j > 0] = \mathbb{E}[-d_j \mid d_j < 0]$$

We show this using the integral definition of conditional expectation. The symmetric disagreement assumption implies $f(x) = f(-x)$ and $\mathbb{P}(d_j > 0) = \mathbb{P}(d_j < 0)$. Consider the right-hand side:

$$\mathbb{E}[-d_j \mid d_j < 0] = \frac{\int_{-\infty}^0 -x f(x)\, dx}{\mathbb{P}(d_j < 0)}.$$

Applying the change of variable $u = -x$, we have $x = -u$ and $dx = -du$. Therefore

$$\frac{\int_{\infty}^0 -(-u) f(-u)(-du)}{\mathbb{P}(d_j < 0)} = \frac{-\int_{\infty}^0 u f(-u) du}{\mathbb{P}(d_j < 0)} = \frac{\int_0^{\infty} u f(-u) du}{\mathbb{P}(d_j < 0)}.$$

Using the symmetry properties $f(-u) = f(u)$ and $\mathbb{P}(d_j < 0) = \mathbb{P}(d_j > 0)$, this is equal to:

$$\frac{\int_0^{\infty} u f(u) du}{\mathbb{P}(d_j > 0)} = \mathbb{E}[d_j \mid d_j > 0].$$

Thus, we have shown $\mathbb{E}[-d_j \mid d_j < 0] = \mathbb{E}[d_j \mid d_j > 0]$, which concludes the proof. $\square$

## C.7 BOOTSTRAP CONFIDENCE INTERVALS

Most metrics in this paper can be summarized by the following two steps:

1. We first calculate a collection of scores $s_1, \ldots, s_N$ at either the event level or the market level, where $N$ denotes the total number of events or markets.

2. We then aggregate these scores into a single statistic via a mapping $T : \mathbb{R}^N \to \mathbb{R}$ (typically the arithmetic mean).

To assess the uncertainty of the resulting point estimate, we construct a nonparametric bootstrap confidence interval (CI) at level $(1 - \alpha)\%$ (DiCiccio & Efron, 1996). Specifically, we form a symmetric interval around the point estimate using bootstrap resampling. A pseudo-code for our implementation is given below in Algorithm 1.

---

**Algorithm 1** Symmetric Nonparametric Bootstrap CI Centered at Point Estimate $\hat{\theta}$

---

**Require:** Data $s_1, \ldots, s_N$; statistic $T(\cdot)$; number of bootstrap replicates $B$; target level $1 - \alpha$;
**Ensure:** Symmetric CI $[\hat{\theta} - h,\ \hat{\theta} + h]$ and achieved bootstrap mass $k/B$.
1: Compute point estimate $\hat{\theta} \leftarrow T(s_1, \ldots, s_N)$.
2: **for** $b = 1$ **to** $B$ **do**
3:   Draw a bootstrap sample $\{s_1^{*(b)}, \ldots, s_N^{*(b)}\}$ by sampling with replacement from $\{s_i\}_{i=1}^N$.
4:   Compute bootstrap re-estimate $\theta^{*(b)} \leftarrow T(s_1^{*(b)}, \ldots, s_N^{*(b)})$.
5: **end for**
6: Compute deviations $d_b \leftarrow \left| \theta^{*(b)} - \hat{\theta} \right|$ for $b = 1, \ldots, B$.
7: Sort $\{d_b\}$ into order statistics $d_{(1)} \leq \cdots \leq d_{(B)}$.
8: $k \leftarrow \lceil (1 - \alpha) B \rceil$.
9: $h \leftarrow d_{(k)}$.
10: **return** Symmetric CI $[\hat{\theta} - h,\ \hat{\theta} + h]$.

---

## C.8 ROBUSTNESS TO EVENT-CATEGORY REBALANCING

To examine whether our results are sensitive to the distribution of event categories, we perform robustness analyses using rebalanced subsets of the evaluation data. The main dataset is heavily skewed toward Sports events; to assess the impact of this imbalance, we construct alternative evaluation sets in which the proportion of Sports events is reduced while still retaining a mix of the other categories. One subset retains Sports as the largest category but reduces its dominance to roughly 50% (Sports: 0.5, Entertainment: 0.15, Politics: 0.15, Other: 0.2). Another subset ensures that all

four categories are equally represented (25% each). All analyses use only submissions on or before 2025–09–10.

Table 6 reports the Brier scores across the original and rebalanced evaluation sets. The relative ordering of the top five models remain unchanged across all distributions, confirming that their comparative performance is robust to event-category composition. As the proportion of Sports events decreases, overall Brier scores improve slightly, suggesting that Sports events are intrinsically more challenging to predict.

Table 6: Brier scores across the original and rebalanced evaluation subsets. Numbers in parentheses denote 95% confidence intervals. The "Moderate" subset contains 50% Sports events, while the "Full" subset has 25% Sports events.

| Forecaster | Original (N=816) | Moderate (N=250) | Full (N=152) |
|---|---|---|---|
| GPT-5[R] | 0.179 ($\pm$0.008) | 0.146 ($\pm$0.013) | 0.128 ($\pm$0.015) |
| Grok 4[R] | 0.186 ($\pm$0.009) | 0.156 ($\pm$0.014) | 0.141 ($\pm$0.018) |
| Claude Sonnet 4[R] (Thinking) | 0.190 ($\pm$0.009) | 0.162 ($\pm$0.016) | 0.148 ($\pm$0.020) |
| Gemini 2.5 Flash[R] (Reasoning) | 0.195 ($\pm$0.009) | 0.165 ($\pm$0.016) | 0.157 ($\pm$0.021) |
| Llama 4 Scout | 0.230 ($\pm$0.011) | 0.202 ($\pm$0.019) | 0.210 ($\pm$0.025) |
| Market Baseline | 0.188 ($\pm$0.008) | 0.164 ($\pm$0.010) | 0.149 ($\pm$0.013) |

As such, these results provide evidence that the conclusions drawn in the main text regarding LLM performance are robust to variations in the event-category distribution.

# D ADDITIONAL EXPERIMENT RESULTS

## D.1 FULL EVALUATION RESULTS

| LLMs | Forecasting Loss | | Calibration Error | | Market Return | |
|---|---|---|---|---|---|---|
| | ↓ **Brier** (95% CI) | **Rank** | ↓ **ECE** | **Rank** | ↑ **Average** (95% CI) | **Rank** |
| GPT-5$^R$ (High) △ | 0.184 (± 0.006) | ① | 0.042 | ④ | 0.943 (± 0.042) | ⑥ |
| GPT-5$^R$ | 0.187 (± 0.005) | ② | 0.044 | ⑥ | 0.890 (± 0.040) | 13 |
| Market Baseline | 0.187 (± 0.006) | ③ | 0.069 | 19 | 0.899 (± 0.043) | 10 |
| GPT-5$^R$ (Minimal) | 0.188 (± 0.006) | ④ | 0.036 | ② | 0.869 (± 0.044) | 19 |
| o3$^R$ | 0.188 (± 0.005) | ⑤ | 0.030 | ① | 0.959 (± 0.109) | ⑤ |
| Grok-4$^R$ △ | 0.189 (± 0.005) | ⑥ | 0.043 | ⑤ | 0.864 (± 0.052) | 21 |
| Grok-3-Mini$^R$ | 0.189 (± 0.006) | 7 | 0.046 | 7 | 0.869 (± 0.043) | 20 |
| GPT-4.1 | 0.192 (± 0.007) | 8 | 0.053 | 10 | 0.907 (± 0.035) | 8 |
| Gemini 2.5 Pro$^R$ | 0.193 (± 0.007) | 9 | 0.061 | 14 | 0.876 (± 0.050) | 17 |
| Claude Opus 4.1$^R$ | 0.193 (± 0.018) | 10 | 0.054 | 11 | 0.982 (± 0.093) | ① |
| Claude Sonnet 4$^R$ △ | 0.194 (± 0.006) | 11 | 0.041 | ③ | 0.909 (± 0.101) | 7 |
| o3-Mini$^R$ | 0.195 (± 0.005) | 12 | 0.046 | 8 | 0.897 (± 0.046) | 12 |
| o4-Mini$^R$ (High) | 0.196 (± 0.006) | 13 | 0.062 | 16 | 0.874 (± 0.040) | 18 |
| Kimi-K2 | 0.197 (± 0.008) | 14 | 0.048 | 9 | 0.966 (± 0.124) | ③ |
| Gemini 2.5 Flash$^R$ △ | 0.197 (± 0.007) | 15 | 0.067 | 17 | 0.883 (± 0.053) | 15 |
| GPT-4o | 0.198 (± 0.007) | 16 | 0.058 | 12 | 0.970 (± 0.104) | ② |
| DeepSeek-V3 | 0.201 (± 0.008) | 17 | 0.061 | 15 | 0.963 (± 0.103) | ④ |
| Gemini 2.5 Flash | 0.203 (± 0.006) | 18 | 0.073 | 20 | 0.859 (± 0.042) | 22 |
| Llama-4-Maverick | 0.208 (± 0.008) | 19 | 0.067 | 18 | 0.904 (± 0.050) | 9 |
| Llama-4-Scout △ | 0.219 (± 0.008) | 20 | 0.060 | 13 | 0.805 (± 0.040) | 24 |
| Gemini 2.0 Flash (Lite) | 0.224 (± 0.013) | 21 | 0.091 | 22 | 0.855 (± 0.074) | 23 |
| Gemini 2.0 Flash | 0.224 (± 0.013) | 22 | 0.079 | 21 | 0.876 (± 0.078) | 16 |
| Qwen3-235B | 0.234 (± 0.007) | 23 | 0.118 | 23 | 0.898 (± 0.111) | 11 |
| DeepSeek-R1$^R$ | 0.303 (± 0.018) | 24 | 0.165 | 24 | 0.884 (± 0.058) | 14 |

Table 7: The full evaluation results for all 23 LLMs (including variants) and the market baseline. △ denotes models selected for presentation in the main text.

## D.2 SHARPE RATIO FOR AVERAGE RETURN

As discussed in Section 3.2 and Table 7, Average Return often exhibits wide confidence intervals, reflecting the high variance of event-level payoffs. In most events, models earn nothing or only marginal gains, while in rare cases they achieve outsized returns. To account for this variability, we complement Average Return with the *Sharpe ratio* (Sharpe, 1998), a standard metric that normalizes expected returns by their volatility. Formally, for asset returns $R_a$, the Sharpe ratio is defined as

$$S_a = \frac{\mathbb{E}[R_a - R_b]}{\sqrt{\mathrm{Var}[R_a - R_b]}}, \tag{8}$$

where $R_b$ denotes a reference risk-free return. In the `Prophet Arena` setting, $R_a$ corresponds to the payoff from each event under the trading strategy of Appendix C.3, and we set $R_b = 1$, the return from abstaining (i.e., keeping the budget unbet). The expectation and variance in Eq. (8) are estimated by the sample mean and variance over $n$ events.

Sharpe ratios for all evaluated models are reported in Table 8, providing a volatility-adjusted comparison of economic performance.

| LLMs | ↑ Sharpe Ratio | Rank | LLMs | ↑ Sharpe Ratio | Rank |
|---|---|---|---|---|---|
| o3-Mini[R] | -0.0036 | ① | Gemini 2.5 Flash | -0.0747 | 12 |
| GPT-5[R] | -0.0074 | ② | Grok-3-Mini[R] | -0.0772 | 13 |
| Gemini 2.5 Flash[R] △ | -0.0100 | ③ | o4-Mini[R] | -0.0868 | 14 |
| Gemini 2.5 Pro[R] | -0.0139 | ④ | DeepSeek-V3 | -0.0952 | 15 |
| o3[R] | -0.0141 | ⑤ | Qwen3-235B | -0.1080 | 16 |
| GPT-5[R] (High) △ | -0.0407 | ⑥ | GPT-4o | -0.1114 | 17 |
| Grok-4[R] △ | -0.0467 | 7 | DeepSeek-R1[R] | -0.1135 | 18 |
| GPT-4.1 | -0.0515 | 8 | GPT-5[R] (Minimal) | -0.1269 | 19 |
| Llama-4-Maverick | -0.0549 | 9 | Gemini 2.0 Flash | -0.1684 | 20 |
| Claude Sonnet 4[R] △ | -0.0574 | 10 | Kimi-K2 | -0.2456 | 21 |
| Llama-4-Scout △ | -0.0599 | 11 | Gemini 2.0 Flash (Lite) | -0.2600 | 22 |

Table 8: **Sharpe ratio performance for all 22 LLMs.** △ denotes models selected for presentation in the main text.

### D.3 PROBABILITY ELICITATION METHODS

Section 2 introduced our default approach: directly prompting an LLM to verbalize the probability that a market resolves to `Yes`. In this experiment, we conduct ablation studies over alternative confidence estimation methods. The goal is twofold: (i) test the robustness of our default prompt against reasonable variations, and (ii) illustrate how `Prophet Arena` can serve as a testbed for comparing black-box confidence estimation methods under forecasting settings.

**Setup.** We evaluate the five representative LLMs from main text on the PROPHET-ARENA-SUBSET-100 dataset. Metrics are Brier and ECE scores. We consider two families of confidence estimation methods (seven variants total):

1. **Verbalized Probability** (Tian et al., 2023)
   - *Prompt variation*: we modify our default prompt to make it: (A) more concise, (B) more verbose, and (C) rewritten by another LLM (Grok 4). Key logistics and formatting instructions are preserved in all variations. These variation prompts are available in Appendix F.4.
   - *Prompt emsemble* (Wightman et al., 2023): for each market, we average the probabilities elicited from the default and all variation prompts (i.e. (A), (B), (C) above).
   - *Bi-direction\** (ours): in addition to eliciting $p_{ij}$ (probability of `Yes`) using default prompt, we also elicit $p_{ij}^{\circ}$ (probability of NO), and calibrate via $\frac{1}{2}(p_{ij} + (1 - p_{ij}^{\circ}))$.
2. **Self-consistency** (Wang et al., 2022)
   - *Unweighted*: instead of directly asking for probability, we repeatedly query the model 10 times for a `Yes`/`No` decision; probability is the fraction of `Yes` answers.
   - *Weighted* (Taubenfeld et al., 2025): we further supplement each `Yes`/`No` with a confidence score.[11] Probabilities are formed by confidence-weighted aggregation.

**Results (see Table 9).** Among the verbalized probability methods, we observe that **all models exhibit strong robustness to prompt variations**. For accuracy, Brier scores for all models vary by less than one standard deviation ($\approx 0.01$). As a result, GPT-5 ranks the highest under all methods, and Gemini 2.5 Flash (Thinking) consistently achieves second place in $5/6$ cases. Prompt ensemble does not lead to substantial improvement, since elicited probabilities are already similar across variations. For calibration, GPT-5 and Llama 4 Scout are consistently the best and the worst models, regardless of the prompting method. Despite slightly larger fluctuations among the ECE scores, **no single method dominate the others for all models** (i.e. achieves the best scores on all columns). Our original Bi-direction\* method improves calibration (over the default) on $4/5$ models, supporting the view that LLMs tend to be overconfident toward `Yes` outcomes.

In contrast, **self-consistency methods result in significantly lower accuracies and yield mixed calibration benefits**. With 10 rollouts, the unweighted variant produces coarse-grained probabilities at a resolution of 0.1, limiting accuracy despite incurring higher compute cost. The weighted variant

---

[11] This confidence reflects uncertainty about the chosen answer, not a direct market probability (e.g., low confidence in `Yes` signals indecision, not belief in NO).

| Method Type / Name | Grok 4 | Gemini 2.5 Flash | Claude Sonnet 4 | GPT-5 | Llama 4 Scout |
|---|---|---|---|---|---|
| *Verbalized Prob* | | | | | |
| Default | 0.186/0.117 | 0.166/0.036 | 0.173/0.046 | **0.165/0.020** | 0.196/0.153 |
| Variation A (Concise) | 0.180/0.102 | 0.172/0.036 | 0.169/0.035 | **0.162/0.021** | 0.192/0.134 |
| Variation B (Verbose) | 0.178/0.124 | 0.166/0.039 | 0.172/0.040 | **0.160/0.028** | 0.199/0.164 |
| Variation C (Rewrite) | 0.176/0.123 | 0.167/0.027 | 0.172/0.039 | **0.159/0.016** | 0.195/0.167 |
| Ensemble | 0.177/0.117 | 0.165/0.032 | 0.170/0.043 | **0.160/0.024** | 0.192/0.142 |
| Bi-direction* | 0.180/0.101 | 0.164/0.031 | 0.165/0.028 | **0.158/0.023** | 0.203/0.140 |
| *Self-consistency* | | | | | |
| Unweighted | 0.238/0.115 | **0.231**/0.110 | 0.241/0.071 | 0.239/**0.071** | 0.267/0.129 |
| Weighted | 0.205/0.091 | 0.201/0.067 | 0.189/0.050 | **0.181/0.046** | 0.214/0.125 |

Table 9: **Evaluation Results for Different Probability Elicitation Methods .** Each cell contains a pair of *Brier↓ / ECE↓* scores. **Bold** denotes the best score for each row, underline for each column.

partially alleviates this by using finer-grained confidence signals, producing noticeable calibration gains but still trailing verbalized methods in accuracy.

To sum up, these results show that (i) LLMs are generally robust against prompt ablations, justifying our default prompt choice, and (ii) `Prophet Arena` provides a natural benchmark for evaluating and contrasting future confidence estimation/calibration methods.

### D.4 REASONING CONSISTENCY OF LLMS

In addition to the primary relative (Brier score) and absolute (Average return) metrics, certain types of forecasting events also enable us to evaluate the consistency – an important component of reasoning – of LLMs. These metrics can be calculated **solely by looking at the potential event outcomes and the LLM probabilities given to them.** Below we give two concrete examples of such consistency metrics.

**Logical chain score.**[12]  Consider the forecasting event *"The bitcoin price by the end of 2026"*, and two of its outcomes are *"(A) The bitcoin price is above \$200,000"* and *"(B) The bitcoin price is above \$220,000"*, respectively. No matter how good an LLM is at predicting the probabilities for (A) & (B), we know that anyone with **consistent reasoning** will give $\mathbb{P}[(A)] \geq \mathbb{P}[(B)]$ since the latter outcome logically implies the former. We denote such a relationship as a **logical chain**, or $(B) \to (A)$. Obviously, this logical chain can contain more than two outcomes, so we call a logical chain $\mathcal{S} = (S_1) \to ... \to (S_T)$ **maximal** whenever it satisfies both:

1. For all $1 \leq t < T$, we have $(S_t) \to (S_{t+1})$,
2. No other outcome $(K) \notin \mathcal{S}$ satisfies $(K) \to (S_1)$ or $(S_T) \to (K)$.

A single event might contain multiple maximal logical chains $\mathcal{S}_1, ..., \mathcal{S}_n$ with lengths $T_1, ..., T_n$. For an LLM with probability $\mathbb{P}[(A)]$ for outcome $(A)$, its **logical chain score** for this event is given by $\frac{1}{n} \sum_{i=1}^{n} score(\mathcal{S}_i)$, where

$$score\left(\mathcal{S}_i = (S_{i1} \to ... \to S_{iT_i})\right) := \frac{1}{T_i - 1} \sum_{j=1}^{T_i-1} \mathbf{1}\{\mathbb{P}[(S_j)] \leq \mathbb{P}[(S_{j+1})]\} \quad (9)$$

with $\mathbf{1}(\cdot)$ being the indicator function. The final logical chain score is then **averaged over all events with at least one chain**. We adopt an LLM-judge (`Gemini-2.5-Flash`) to automatically detect the maximal logical chains for all our events.

**Mutually exclusive score.** In certain forecasting events, the potential outcomes are **mutually exclusive**, meaning that exactly one outcome can occur. For example, in the event *"Who will win the NBA championship in 2026?"*, the possible outcomes could be the teams, where only one team can win. A **maximal set of mutually exclusive outcomes** is a set where

1. each outcome is distinct and all outcomes are mutually exclusive,
2. the event will resolve to one and only one outcome in the set.

---

[12]This is a placeholder name. Feel free to suggest a better one.

If such a maximal set $\mathcal{S} := \{(S_1), ..., (S_m)\}$ with size $m$ exists for a forecasting event, we can calculate the **mutually exclusive score** at this event as:

$$score_{\text{ME}} = \mathbf{1}\left\{\sum_{i=1}^{m} \mathbb{P}[(S_i)] = 1\right\} \tag{10}$$

(In practice, we allow the sum to deviate slightly from 1 with some tolerance level $\epsilon$).

We evaluate this score over all events where mutually exclusive outcomes are defined. The identification of maximal set is performed using the same LLM-judge (`Gemini-2.5-Flash`).

| LLMs | Mutually Exclusive Consistency | Logical Chain Consistency |
|---|---|---|
| DeepSeek-R1[R] | 0.996 | 0.987 |
| o4-Mini[R] | 0.995 | 0.998 |
| GPT-5[R] (High) | 0.995 | 0.999 |
| Gemini 2.5 Flash[R] | 0.995 | 0.997 |
| Gemini 2.5 Pro[R] | 0.995 | 0.995 |
| Claude Sonnet 4[R] | 0.995 | 0.987 |
| DeepSeek-V3 | 0.994 | 0.973 |
| Grok-4[R] | 0.994 | 0.994 |
| GPT-4.1 | 0.994 | 0.973 |
| Llama 4 Maverick | 0.994 | 0.981 |
| Llama 4 Scout | 0.994 | 0.979 |
| Grok-3-Mini[R] | 0.994 | 0.901 |
| Qwen3-235B | 0.994 | 0.988 |
| GPT-4o | 0.994 | 0.930 |
| GPT-5[R] (Minimal) | 0.994 | 0.962 |
| GPT-5[R] (Medium) | 0.994 | 0.990 |
| o3-Mini[R] | 0.994 | 0.930 |
| Gemini 2.0 Flash | 0.994 | 0.990 |
| Kimi-K2 | 0.993 | 0.984 |
| o3[R] | 0.993 | 0.996 |
| Gemini 2.0 Flash (Lite) | 0.993 | 0.911 |

Table 10: **Consistency scores for the 22 LLMs.** Most LLMs evaluated have exhibited excellent performances in both consistency metrics, indicating their mature logical reasoning skills.

### D.5 Variability in Model Forecasts Despite Identical Inputs

While all tested LLMs receive the same market data and news sources, they differ substantially in how they combine and weigh each piece of information, resulting in diverse prediction patterns. Fig. 8 shows averaged pairwise differences in event predictions, and the generally large differences across models suggest that LLMs reason about events in fundamentally different ways, even when given identical inputs.

Interestingly, even within model families, clustering patterns also vary notably. For example, the GPT-5 variants produce relatively homogeneous predictions, reflecting similar training and reasoning capabilities. In contrast, Llama 4 Maverick and Scout, despite belonging to the same model family, exhibit the second largest average pairwise difference (0.43). This divergence illustrates that sharing a model family does not guarantee similar predictions, nor does it imply identical reasoning processes.

### D.6 Conservatism of Model Predictions Relative to Market-Implied Predictions

Figure 9 shows that all tested models exhibit a conservative tendency, assigning significantly lower probabilities than markets on numerous events. The magnitude of the conservative tendency varies by model, but the general trends Section 4.1.3 discussed in observed persist.

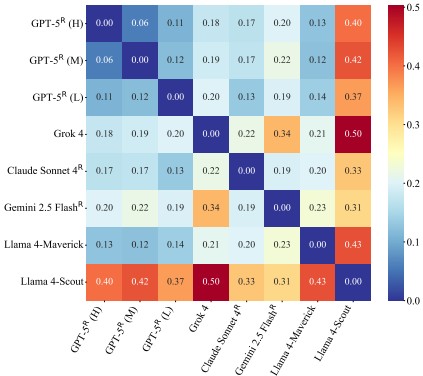

Figure 8: **Pairwise divergence of model predictions across tasks.** Heatmap shows the average $L_2$ distance between probabilistic predictions of each model pair. GPT-5 H, M, L labels represent high, medium, and minimal reasoning efforts, as explained by Appendix B.6.

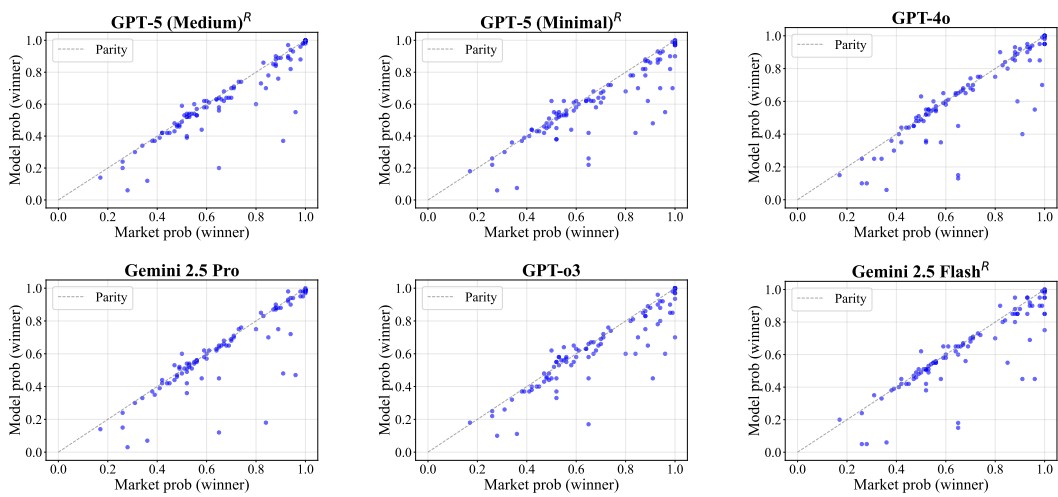

Figure 9: **LLM prediction probabilities compared to market baselines.** Scatter plots show the relationship between market-implied probabilities and model-assigned probabilities. Points on the diagonal line correspond to events where the model prediction matches the market probability.

## D.7 EVALUATING REASONING USING AN LLM-AS-A-JUDGE FRAMEWORK

To directly evaluate LLMs reasoning processes, we employ an LLM-as-a-judge framework (Zheng et al., 2023) to assess the soundness of their reasoning methodologies. Our evaluation proceeds in two stages: (1) eliciting explicit reasoning from the prediction model, and (2) systematically evaluating this reasoning with an independent evaluator.

The assessment framework encompasses five critical dimensions, each scored on an 1-5 point scale (where 1 and 5 indicate poor and excellent performance, respectively):

1. **Source selection**: Assessment of how effectively models incorporate provided sources and the reliability of their source selection criteria.

2. **Evidence extraction from selected sources** : Evaluation of the model's ability to extract relevant evidence from sources and demonstrate sophisticated interpretation beyond surface-level analysis.

3. **Reasoning synthesis**: Analysis of how extracted evidence is integrated into coherent justifications, including the model's approach to combining and weighting disparate pieces of evidence.

4. **Reasoning-to-prediction**: Assessment of how effectively the reasoning process is translated into the final probabilistic prediction.

5. **Recognition of prediction uncertainty**: Examination of the model's capacity to identify and appropriately account for uncertainties and potential counterarguments within their analysis.

To enhance evaluation reliability, we incorporate a human expert-assessed reference evaluation of an external event not included in the dataset, which serves as a grounding benchmark. Additionally, we lower the temperature setting to 0 for the LLM judge (Claude Sonnet 4) to improve response consistency. We include the full prompt in Appendix F.2.

We include the full table of reasoning evaluations with a wider range of LLMs below. The same trends observed in Section 4.3 are evident here: models reach near-parity in source utilization, evidence extraction, and uncertainty analysis, while exhibiting substantial disparities in reasoning synthesis and reasoning-to-prediction alignment. These latter dimensions are the primary drivers of differences in overall predictive performance. As such, the findings further suggest that the development of future prediction agents should prioritize advances in higher-order reasoning and the alignment of reasoning with probabilistic forecasts, rather than focusing on marginal improvements in retrieval or evidence handling.

| LLM | Sources | Evidence | Reas. Synth. | Align. | Uncert. | Average Score |
|---|---|---|---|---|---|---|
| GPT-5$^R$ (High) | 3.69 | 3.66 | **4.14** | **3.97** | 3.94 | **3.88** |
| O3 | **3.71** | **3.74** | 3.93 | 3.78 | 3.87 | 3.81 |
| Gemini 2.5 Pro | 3.70 | 3.69 | 3.39 | 3.92 | **3.95** | 3.73 |
| GPT-5$^R$ (Medium) | 3.69 | 3.65 | 3.69 | 3.66 | 3.94 | 3.73 |
| GPT-5$^R$ (Minimal) | 3.69 | 3.64 | 3.26 | 3.58 | 3.90 | 3.61 |
| Gemini 2.5 Flash$^R$ | 3.57 | 3.66 | 3.19 | 3.67 | 3.74 | 3.57 |
| Grok 4 | 3.40 | 3.51 | 3.33 | 3.48 | 3.66 | 3.48 |
| Claude Sonnet 4$^R$ | 3.53 | 3.47 | 2.93 | 3.39 | 3.75 | 3.41 |
| Llama 4 Maverick | 3.14 | 3.29 | 2.43 | 2.14 | 3.14 | 2.83 |
| GPT-4o | 3.07 | 2.99 | 2.32 | 2.59 | 2.96 | 2.79 |
| Llama 4 Scout | 2.97 | 2.88 | 2.29 | 2.37 | 2.87 | 2.68 |

Table 11: **Full table on LLM performance on reasoning evaluation criteria across dataset events.** Each dimension is scored on a standardized 5-point scale, where 1 and 5 indicate poor and excellent performance, respectively. Average scores are presented for each model, with **bold** values indicating the best-performing model for each criterion. Models are ordered by descending overall average score.

### D.7.1 Validating LLM-as-a-Judge via Human Ratings

To validate the LLM-as-a-judge system and check for alignment bias, we conducted a blinded human adjudication study. We randomly sampled 170 events with LLM-generated reasoning traces and hid both the model identity and LLM judge outputs from human raters. Each reasoning trace was independently scored by humans on a 1-5 scale across the five rubric dimensions (Sources Used, Evidence Extracted, Combination & Weighting, Uncertainties & Counterpoints, and Mapping to Final Probabilities). The rubric provided was identical to that given to the LLMs.

Overall, we find a high degree of concordance between human ratings and LLM-as-a-judge scores. As shown in Table 12, mean absolute differences between human and LLM ratings ranged from 0.37 to 0.44, with an average difference of 0.42, and variances between 0.27 and 0.36. These values are strikingly low given the 1-5 rating scale: all differences are, on average, substantially smaller than 1, meaning that the LLM judges score deviates from human judgment by less than half a point. Moreover, across all rubric categories, over 94% of human ratings were within 1 point of the LLMs score as seen in Fig. 10, underscoring the tight alignment between automated and human evaluation.

### D.7.2 (Lack of) Effect of Scaffolding Prompt on Prediction

Table 13 reports the differences in Brier score between the predictions with the reasoning scaffolding prompt and the predictions with the default, non-scaffolding configurations. Overall, the differences are insignificant for all models, indicating that the probabilistic forecasts remain largely stable regardless of the prompt type. This suggests that the enhanced reasoning elicitation prompt

| Rubric Category | Mean Absolute Difference (0-4) (Variance) |
|---|---|
| Sources Used | 0.43 (0.30) |
| Evidence Extracted | 0.42 (0.27) |
| Combination Weighting | 0.43 (0.36) |
| Uncertainties Counterpoints | 0.44 (0.28) |
| Mapping To Final Probabilities | 0.37 (0.29) |

Table 12: Human ratings for rubric categories, with variance shown in parentheses. Mean absolute difference is calculated as the average $|$ human score $-$ LLM score $|$ across the 170 events. Differences are on a 0-4 scale.

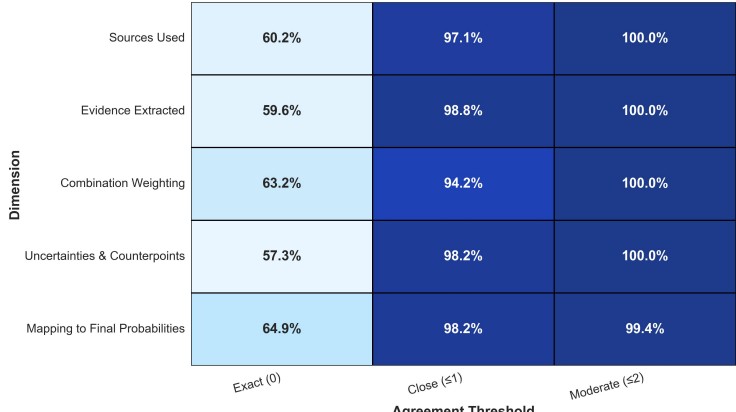

Figure 10: Agreement matrix showing the percentage of human ratings that lie within exact, moderate ($\pm1$), and larger ($\pm2$) thresholds of the LLM-judge ratings for each rubric dimension.

did not meaningfully improve prediction performance, implying that the prompt primarily serves as a structured summary rather than enhancing the models underlying predictive capabilities.

| Model | Brier Difference |
|---|---|
| GPT-5[R] (High) | -0.0012 |
| GPT-5[R] (Medium) | 0.0011 |
| GPT-5[R] (Minimal) | 0.0017 |
| o3[R] | -0.0018 |
| Gemini 2.5 Pro[R] | -0.0080 |
| Gemini 2.5 Flash[R] | -0.0001 |
| Llama 4 Maverick | 0.0095 |
| Llama 4 Scout | 0.0138 |
| Grok-4[R] | -0.0186 |
| Claude Sonnet 4[R] | -0.0005 |
| DeepSeek-R1[R] | 0.0057 |
| GPT-4o | 0.0053 |

Table 13: Brier differences between the reasoning scaffolding prompts and non-scaffolding configurations.

## D.8 KNOWLEDGE INTERNALIZATION

### D.8.1 KNOWLEDGE INTERNALIZATION EVENTS

We sample 100 events from Kalshi (Inc., 2025), with market close time before October 2023 (i.e., before all model knowledge cutoff dates. Note that despite that many popular events on Kalshi are *Sports* events as of August 2025, sports betting was not legal on Kalshi until 2024 (Wilmot,

2025). The sampled past events span the categories available on the platform during that period and exhibit differing levels of *temporal granularity*. Some events target a specific timestamp or date (e.g., *NASDAQ price on August 20, 2023*), while others are coarser, period-level questions without a single focal timestamp (e.g., *Will WTI crude oil prices decrease in Q2 2023?*).

Figure 11: Event distribution for memorization experiments

### D.8.2 KNOWLEDGE INTERNALIZATION PROMPTS

We use three complementary prompts to test models' knowledge internalization.

1. **Prediction prompt (no sources)**: the original forecasting-style prompt used in `Prophet Arena` without sources context (Appendix F.3.2). Although framed as a forward-looking prediction, all of these past events are in fact already represented in the model's training data. Success in these cases indicates that the model can implicitly recognize the event as historical and draw on its internalized knowledge to answer correctly. Failure, in contrast, highlights a gap between memorization and reasoning: the model may know the fact but still treat the prompt purely as a forecasting task, leading to mis-recall

2. **Prediction prompt with sources**: the same forecasting prompt with an additional block (Appendix F.3.3), but augmented with event-specific sources. In this setting, the model is no longer reasoning only from internalized knowledge: it must integrate retrieved evidence with what it already "remembers." This setup tests whether the model can align its internal recall with external evidence, and whether retrieval corrects, reinforces, or conflicts with its memorized knowledge.

3. **Recall prompt**: a specialized prompt that explicitly frames the task as recalling a past outcome (Appendix F.3.1). This isolates the models internalized knowledge, revealing whether it has retained coarse or precise details about prior events.

### D.9 HOW GOOD ARE LLMS AT FINDING SOURCES?

We evaluate models' source-finding ability on the set of *past* events (Appendix D.8.1), using the search prompt. Because these events have resolved, accurate information is publicly available; competent search should surface the correct evidence and hence, on average, improve the quality of the models' recall. In Fig. 12b, we observe that the addition of sources improve LLM recall quality in certain events but not others. In fact, the figure shows that the events hardest to recall often becomes worse from the source-augmented variant. One likely explanation is that retrieved sources, while factual, introduce noise or extraneous details that interfere with the models internalized recall and reasoning. Events requiring precise, fine-grained recall (e.g., financial indicators or political approval ratings) tend to suffer when sources are added, likely because the retrieved evidence contains multiple overlapping numbers, dates, or conditions that confuse the model. By contrast, more salient and broadly covered events (e.g., major stock index movements or entertainment outcomes) generally improve with sources, since the found sources likely have lower variance and higher accuracy.

This demonstrates that even on past events that have closed, LLM searchers do not yet possess the ability to accurately pinpoint the most useful sources for finer-granularity events.

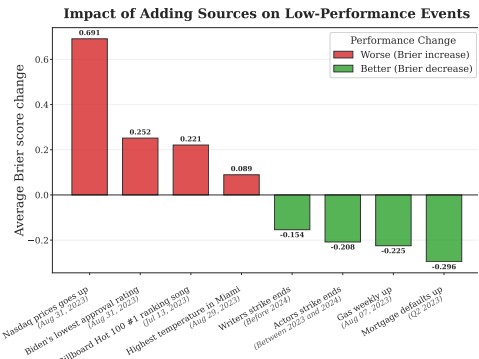 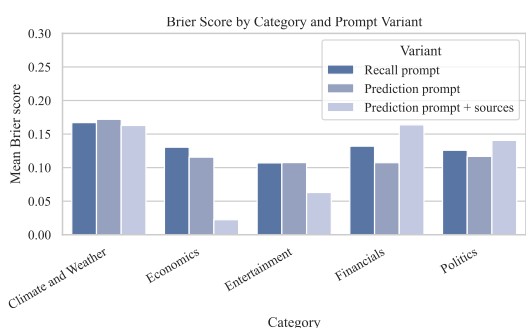

Figure 12: **Impact of adding sources to context on past events.** The low-performance events refer to past events that the models have worst recall on. The average Brier score change is computed by evaluating the difference between when models are provided with searched sources vs. when models are solely recalling based on memory.

# E  CASE STUDIES

## E.1  DIFFERENCES IN PREDICTION WITH THE SAME INFORMATION

In the following example, the LLMs predict on the event, the Club World Cup soccer game between Real Madrid and Al Hilal SFC, scheduled for Jun 18, 2025. The sources and market snapshots that LLMs receive are also the same, detailed in Table 14. The market prices were around (74% Madrid, 15% tie, 14% Al Hilal). Despite given the same market and source information, models showed differing rationales.

GPT-4o for example, explicitly adjusted its prediction to account for the possibility of a draw, predicting (70% Madrid, 20% tie, 10% Al Hilal), noting:

> "Sources universally favor Real Madrid, citing their superior quality, historical victory, and Al Hilals weakened state without Mitrovi. Market predictions strongly back Real Madrid, **but I adjusted the probabilities slightly to account for Real Madrids lineup challenges and adaptation under a new coach. The uncertain defensive setup and potential absence of Mbapp slightly raise the probability of a tie.**

Gemini 2.5 Flash, in contrast, acknowledged the sources that suggest a higher chance of a tie, but nevertheless decides to place a higher weight on Madrid win (79% Madrid, 13% tie, 8% Al Hilal). It reasons that:

> "I heavily weighed the consistent expert analysis, betting odds, and market data, all strongly favoring Real Madrid. The probability distribution reflects Real Madrid's clear dominance and superior quality, further amplified by the critical injury to Al Hilal's top scorer. **Although Real Madrid faces minor lineup challenges and is adapting to a new tactical approach, these factors are significantly outweighed by Al Hilal's weakened offensive capabilities and the overall disparity in team strength.**"

These fine-grained distinctions are not rarely found in our collected data. They reflect how models reason under uncertainty, and often contributes to the differences in model performances.

## E.2  EVENT RECALL IS APPROXIMATE, NOT PRECISE

Below is the model output of Gemini 2.5 Flash on the event *Billboard Hot 100 #1, Jul 13, 2023?*, from the Recall prompt in Appendix D.8.2.

Table 14: Sources used for Real Madrid vs. Al Hilal SFC match.

| Source | Summary | Title |
|---|---|---|
| Sporting News | Real Madrid is heavily favored to win against Al Hilal, with betting odds reflecting their dominance despite potential lineup challenges. | Real Madrid vs Al Hilal prediction, odds, betting tips and best bets for Club World Cup final |
| PokerStars Sports | Betting odds indicate a strong expectation for a Real Madrid victory, with a significant likelihood of multiple goals being scored in the match. | Real Madrid v Al-Hilal Betting Odds — PokerStars Sports |
| Sporting News (India) | Analysts predict a 3–1 victory for Real Madrid, citing Al Hilal's reliance on penalties in previous matches and Madrid's superior quality. | Real Madrid vs Al Hilal prediction, odds, betting tips and best bets for Club World Cup final |
| BetsLoaded | Real Madrid is predicted to win against Al Hilal, with recent form and head-to-head statistics favoring the Spanish club. | Real Madrid vs Al Hilal Saudi FC Prediction, Betting Tips (18 June 2025) |
| El País | Under Xabi Alonso, Real Madrid is striving to establish a new identity with a focus on high-pressure play, though the team is still adapting to this approach. | El Real Madrid de Xabi busca nueva identidad en Miami: 'Empieza el rock and roll' |
| Sky Sports | Historical data shows Real Madrid's previous victory over Al Hilal, suggesting a favorable outcome for the Spanish team in the upcoming match. | Form and head to head stats Real Madrid vs Al-Hilal |
| AS (Diario AS) | Al Hilal's top scorer, Aleksandar Mitrović, will miss the match against Real Madrid due to a muscle injury, significantly weakening their offensive capabilities. | Al Hilal pierde a Mitrovic |
| AS (Diario AS) | Real Madrid's potential lineup against Al Hilal may see Rodrygo replacing the ill Mbappé, with new signings Alexander-Arnold and Huijsen expected to start in defense. | Alineación posible del Real Madrid contra Al Hilal en el Mundial de Clubes |
| AS (Diario AS) | Real Madrid, under new coach Xabi Alonso, faces defensive challenges due to injuries and is uncertain about its tactical formation ahead of the match against Al Hilal. | Nueva era y nueva defensa |
| Reuters | Kylian Mbappé is doubtful for Real Madrid's match against Al Hilal due to a high fever, potentially leaving the team without a recognized center forward. | Mbappe doubtful for Real Madrid's Club World Cup opener against Al-Hilal |

```
Gemini 2.5 Flash Model Output

Event: TOPSONG-23JUL13
Q: Billboard Hot 100 #1, Jul 13, 2023?

Probabilities:
  - Vampire (Olivia Rodrigo): 1.0
  - Last Night (Morgan Wallen): 0.0
  - Fast Car (Luke Combs): 0.0

Rationale:
  I recall that Olivia Rodrigo's 'Vampire' debuted at
  number one on the Billboard Hot 100 for the chart
  dated July 15, 2023, which corresponds to the
```

```
   July 13, 2023, reference in the event.
   It displaced Morgan Wallen's 'Last Night'.

Recall Assessment:
  - Recognized event: True
  - Evidence:
    1. Olivia Rodrigo's 'Vampire' debuted at #1
       on July 15, 2023.
    2. The song accrued 29.1M streams and sold
       25.7k copies in its first week.
  - Recalled outcome: Vampire by Olivia Rodrigo
```

### E.3 CASE STUDY: BITCOIN PRICE AT 9 AM EDT

Two identical Bitcoin price prediction tasks, separated by five days (July 4 vs. July 9, 2025), illustrate the differences that source quality can have on an LLM's ability to form quality predictions. While the prediction task is essentially identical, for the July 4th event, including sources in the prompt **worsened** prediction accuracy, whereas for the July 9th event, source inclusion **substantially improved** predictions (see App. E.3.1 for the detailed figure). Full source lists for both events are also available in App. E.3.2. Despite the July 4th source list have greater platform diversity (mainstream media, AI-generated predictions, algorithmic forecast bots), it suffers from quality inconsistencies and methodological fragmentation. Numerous questionable forecast bots (e.g., priceforecastbot.com, cryptopredictions.com) produce extremely wide-ranged predictions ($82,822 - $121,797), which works to generate noise rather than useful signals. Similarly, dubious sources like ChatGPT-based predictions further dilute analytical credibility. In contrast, the July 9th dataset exhibits stronger coherence and practical utility, with predictions converging around $115,000 - $125,000 from crypto-specialized platforms (CoinEdition, CoinDCX, Quickex.io, CoinCu) using consistent technical analysis.

Thus, source competence and reliability are critical for enabling LLMs to generate accurate forecasts. As such, these findings show that feeding in mass amounts of information to LLMs does not necessarily enhance prediction quality.

#### E.3.1 BRIER SCORE COMPARISONS

Fig. 13 illustrates the differences in Brier score when sources were added to the prompt for both July 4th and July 9th events. While for the July 4th event, sources significantly improved the prediction quality, for the July 9th event, it worsened the prediction quality.

#### E.3.2 SOURCE LISTS

Below, we provide the complete source lists for both events. Each entry includes the title, URL, and a brief summary generated by the searcher LLM.

```
July 4 Source List

[
  {
    'title': 'Bitcoin (BTC) Price Prediction 2025-2040',
    'url': 'https://changelly.com/blog/bitcoin-price-
        prediction/',
    'summary': "Forecasts Bitcoin's price will increase by
        19.78%, reaching $130,978.70 by July 5, 2025.
        Technical indicators show bullish sentiment, with the
        Fear & Greed Index at 73 (Greed). 16 out of 30 green
        days in the last month."
  },
  {
```

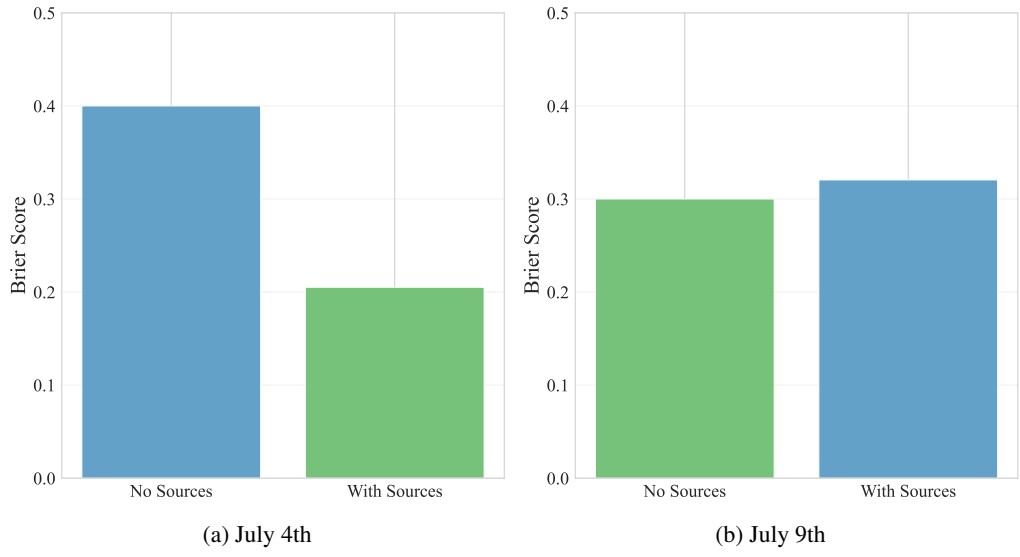

Figure 13: Brier Scores with and without sources in the prompt for July 4th (left) and July 9th (right) Bitcoin events. Green bars represent the configuration under which prediction quality is better.

```
        'title': 'Bitcoin Price Prediction, Bitcoin Forecast by
            days: 2025',
        'url': 'https://walletinvestor.com/forecast/bitcoin-
            prediction-data',
        'summary': "Provides daily BTC price forecasts for July
            2025. Example: July 15 prediction is $102,206 (range
            $94,066 $110,224). Based on historical data and
            market trends."
    },
    {
        'title': 'How High Can Bitcoin (BTC) Soar On July 4,
            2025?',
        'url': 'https://thebittimes.com/how-high-can-bitcoin-btc-
            soar-on-july-4-2025-tbt117327.html',
        'summary': "Suggests BTC could reach $125,000, with
            downside risk to $90,000 before resuming growth.
            Based on 50-day EMA analysis."
    },
    {
        'title': 'Bitcoin Price Prediction: 2025, 2030, 2040',
        'url': 'https://ambcrypto.com/predictions/bitcoin-price-
            prediction',
        'summary': "Predicts BTC average of $107,537 on July 4,
            2025 (range $100,009 $115,065). Indicates steady
            upward trend from current market conditions."
    },
    {
        'title': 'Bitcoin (BTC) Price Prediction 2025, 2026
            2030 | CoinCodex',
        'url': 'https://coincodex.com/crypto/bitcoin/price-
            prediction',
```

```
        'summary': "Forecasts BTC to rise 12.51% to $118,009 by
            July 4, 2025. 17/30 green days, 3.8% volatility. Fear
             & Greed Index at neutral."
    },
    {
      'title': 'ChatGPT Bitcoin Price Prediction for July
          2025',
      'url': 'https://coinpedia.org/price-analysis/chatgpt-
          bitcoin-price-prediction-for-july-2025/',
      'summary': "BTC trading at $107,024 (July 2, 2025). RSI
          and Bollinger Bands show BTC at a critical inflection
           point. Potential breakout with volatility risk."
    },
    {
      'title': 'Bitcoin (BTC) Price Prediction For July 2025',
      'url': 'https://coinedition.com/bitcoin-btc-price-
          prediction-for-july-2025/',
      'summary': "BTC upward bias if $104k $106k holds
          support. Breakout above $110k could lead to $114.5
          k $125k. RSI and MACD support bullish view."
    },
    {
      'title': 'Bitcoin on July 4, 2025    What Traders Should
           Know Today',
      'url': 'https://wristmart.in/bitcoin-on-july-4-2025/',
      'summary': "BTC testing $70k resistance. Breakout could
          spark rally; rejection may cause pullback to $68.2k.
          Includes RSI, MA, and support/resistance zones."
    },
    {
      'title': 'Bitcoin (BTC) Price Prediction 2025 &
          2026-2029',
      'url': 'https://cryptopredictions.com/bitcoin/',
      'summary': "Predicts July 2025 BTC average $97,438 (range
          $82,822 $121,797). Suggests possible correction,
           11.27% from prior months."
    },
    {
      'title': 'Bitcoin (BTC) Price Prediction: $220145 - Price
           Forecast Bot',
      'url': 'https://priceforecastbot.com/coins/bitcoin-price-
          prediction.html',
      'summary': "Forecasts 2025 BTC range: $75,588 $125,981.
          Average prediction $100,785. Based on historical
          data and market analysis."
    },
    {
      'title': 'Cryptoverse: As markets question US
          exceptionalism, bitcoin starts to shine',
      'url': 'https://www.reuters.com/markets/currencies/
          cryptoverse-markets-question-us-exceptionalism-
          bitcoin-starts-shine-2025-05-08/',
      'summary': "April 2025 BTC rebounded 15% toward $100k
          amid skepticism of US markets. Analysts see potential
           rally to $120k in Q2 2025 as investors hedge."
    }
  ]
```

```
July 9 2025 Source List

[
  {
    'title': 'Bitcoin Price Prediction - BTC Forecast 2025,
        2026, 2030',
    'url': 'https://altpricer.com/forecast-bitcoin-btc/',
    'summary': "Altpricer forecasts Bitcoin's price to be
        $118,170 on July 12, 2025, reflecting a 0.25%
        increase from the previous day. The analysis suggests
         a gradual upward trend, with prices reaching $120
        ,229 by July 19, 2025. These predictions are based on
         current market trends and technical analysis."
  },
  {
    'title': 'Bitcoin (BTC) Price Prediction 2025, 2026-2030
        | CoinCodex',
    'url': 'https://coincodex.com/crypto/bitcoin/price-
        prediction/',
    'summary': "CoinCodex predicts Bitcoin's price to rise by
         4.76% to $123,274 by August 10, 2025. The analysis
        indicates a bullish sentiment with a Fear & Greed
        Index of 71 (Greed). It also reports that Bitcoin
        recorded 17 out of 30 green days with 2.14% price
        volatility over the last 30 days."
  },
  {
    'title': 'Bitcoin price prediction for July 2025 |
        Quickex.io',
    'url': 'https://quickex.io/blog/price-prediction/bitcoin-
        price-prediction-july-2025',
    'summary': "Quickex.io reports that a 'bullish flag'
        pattern is forming on Bitcoin's chart, suggesting
        potential for new highs around $115,000 by mid-July.
        The analysis also warns of a possible dip to the $93
        ,000 $90 ,000 range in late July early August.
        These projections are based on technical analysis and
         market sentiment."
  },
  {
    'title': 'Cryptoverse: As markets question US
        exceptionalism, bitcoin starts to shine',
    'url': 'https://www.reuters.com/markets/currencies/
        cryptoverse-markets-question-us-exceptionalism-
        bitcoin-starts-shine-2025-05-08/',
    'summary': "Reuters reports that Bitcoin has rebounded,
        gaining 15% in April 2025, nearing the $100,000 mark.
         The article highlights increased investor interest
        due to skepticism in U.S. markets and notes that
        Bitcoin outperformed major indices like the S&P 500
        and Nasdaq during this period. Analysts suggest
        Bitcoin could reach $120,000 in Q2 2025."
  },
  {
    'title': 'Bitcoin (BTC) Price Prediction for July 12',
    'url': 'https://coinedition.com/bitcoin-btc-price-
        prediction-for-july-12-2025/',
```

```
      'summary': "This article reports that Bitcoin has broken
          through resistance to reach $118,000, its highest
          level since April, driven by ETF inflows and
          institutional interest. Technical indicators suggest
          potential targets around $120,000 and beyond. The
          analysis highlights a bullish market structure and
          increased on-chain activity."
    },
    {
      'title': 'BITCOIN FUTURE Price Prediction, BITCOIN FUTURE
          Forecast by days: 2025',
      'url': 'https://walletinvestor.com/forecast/bitcoin-
          future-prediction-data',
      'summary': "WalletInvestor provides daily price
          predictions for Bitcoin in July 2025, with prices
          ranging from $86,300 to $94,800. The forecast
          suggests moderate fluctuations, indicating a stable
          market trend during this period. These projections
          are based on historical data and market analysis."
    },
    {
      'title': 'Bitcoin (BTC) Price Prediction 2025-2040',
      'url': 'https://changelly.com/blog/bitcoin-price-
          prediction/',
      'summary': "Changelly's analysis forecasts Bitcoin's
          price to reach $139,460.44 by July 9, 2025,
          indicating a 28.82% increase. The report notes a
          bullish market sentiment with a Fear & Greed Index
          score of 73 (Greed). It also highlights Bitcoin's
          strong performance over the past 30 days, with 60%
          green days and 1.89% price volatility."
    },
    {
      'title': 'Bitcoin (BTC) Price Prediction Up To $1
          ,672,861.46 | BTC Forecast',
      'url': 'https://coincu.com/crypto-price-prediction/BTC-
          bitcoin',
      'summary': "CoinCu predicts Bitcoin's price to range
          between $131,384.83 and $150,792.56 in July 2025. The
           analysis indicates potential for significant growth,
           with prices possibly reaching new highs. These
          projections are based on market trends and investor
          sentiment."
    },
    {
      'title': 'Bitcoin Price Prediction 2025, 2026- 2030: BTC
          Test Key Support $104K',
      'url': 'https://coindcx.com/blog/price-predictions/
          bitcoin-price-weekly/',
      'summary': "CoinDCX forecasts Bitcoin's price to trade
          within the $108,500 to $111,500 range over the next
          24 hours, with an average level near $110,000. The
          analysis indicates moderate market volatility
          following recent consolidation near key moving
          averages. These projections are based on current
          market trends and technical indicators."
    },
```

```
{
  'title': 'Bitcoin (BTC) Price Prediction 2025 - 2030 -
      How Will It Perform?',
  'url': 'https://cryptonews.com/news/bitcoin-price-
      prediction.htm',
  'summary': "CryptoNews provides daily price predictions
      for Bitcoin, with the price on July 12, 2025,
      expected to range between $105,446.51 and $108
      ,974.77. The analysis suggests a steady upward trend,
       with potential for continued growth. These forecasts
       are based on historical data and market analysis."
},
{
  'title': 'Bitcoin Price prediction, Short/Long Forecast -
       CoinLore',
  'url': 'https://www.coinlore.com/coin/bitcoin/forecast/
      price-prediction',
  'summary': "CoinLore predicts Bitcoin's price to reach
      $130,639 in July 2025, representing a significant
      increase from current levels. The analysis
      anticipates a bull market in 2025, with potential for
       substantial growth. These projections are based on
      historical data and market trends."
}
]
```

# F PROMPTS

## F.1 PROPHET ARENA PIPELINE PROMPTS

### F.1.1 SEARCH PROMPT

```
Search Prompt

Find exactly 10 recent, relevant sources that can help
    predict the outcome of the event: {event.title}
This is a future event that you are collecting sources for.
    Note that the current time is {datetime.now().strftime("%
    Y-%m-%d %H:%M:%S")} and the event closes at {event.
    close_time.strftime("%Y-%m-%d %H:%M:%S")}.
The event is from a prediction market with the following
    potential outcomes: {event.markets}

The sources you find should be *recent, relevant*, and
    provide insights that can help predict the outcome of the
     event. Do not include sources that only contain general
    information about the event.
For each of the 10 relevant sources you will collect, return:
1. The full URL of the source.
2. The title of the source in *English*. If it is in another
    language, translate it to English. (if not available,
    provide a one sentence English summary as the title).
3. A 3 sentence summary in English of the *key* insights from
     that source.

Respond in the following JSON format:

[
{{
    "source": "https://example.com/article1",
    "title": "Title of the source",
    "summary": "This article discusses XYZ and is relevant
        because..."
}},
...
]
```

### F.1.2 PREDICTION PROMPT

```
Prediction Prompt

You are an AI assistant specialized in analyzing and
    predicting real-world events.
You have deep expertise in predicting the outcome of the
    event: "{event_title}"

Note that this event occurs in the future. You will be given
    a list of sources with their summaries, rankings, and
    expert comments.
Based on these collected sources, your goal is to extract
    meaningful insights and provide well-reasoned predictions
     based on the given data.
```

```
You will be predicting the probability (as a float value from
    0 to 1) of ONLY the following possible outcomes:
{market_list_str}

IMPORTANT CONSTRAINTS:
1. You MUST ONLY provide probabilities for the exact possible
    outcomes listed above
2. Do NOT create or invent any additional outcomes
3. Use exactly the same outcome names as provided (case-
    sensitive)
4. Ensure all probabilities are between 0 and 1

Your response MUST be in JSON format with the following
    structure:
```json
{{
    "rationale": "<text_explaining_your_rationale>",
    "probabilities": {{
        {json_example}
    }}
}}
```

In the rationale section of your response, please provide a
    short, concise, 3 sentence rationale that explains:
- How you weighed different pieces of information
- Your reasoning for the probability distribution you
    assigned
- Any key factors or uncertainties you considered

Note: Market data can provide insights into the current
    consensus of the market influenced by traders of various
    beliefs and private information. However, you should not
    rely on market data alone to make your prediction.
Please consider both the market data and the information
    sources to help you make a well-calibrated prediction.

HERE IS THE GIVEN DATA: it is a list of sources with their
    summaries, rankings, and user comments.
The smaller the ranking number, the more you should weight
    the source in your prediction.
{sources}

CURRENT ONLINE TRADING DATA:
You also have access to the predicted outcome probability (
    last trading price of each outcome turned out to be yes)
    from a popular prediction market at the moment of your
    prediction:
{market_statistics}
```

## F.2 EVALUATING REASONING USING AN LLM-AS-A-JUDGE FRAMEWORK

```
Reasoning Evaluation Prompt

1. Sources Used (Citations, Attribution & Reliability)

**5 - Exceptional:** Every single fact tied to a *direct,
   authoritative* source. Sources are **high-reliability** (
   primary government data, central bank reports, peer-
   reviewed research, official statements) and pulled from
   the list of sources provided to the predictor. Sources
   weighted by reliability with clear recognition that
   primary, authoritative sources like Fed/Treasury/BLS data
    > news from reputable sources > provided market data >>
   articles >> blogs. Zero broken links, zero vague
   attributions. Connection between the rationale and the
   sources is very clear.

**4 - Good:** All major claims properly sourced with mostly
   high-reliability sources dominating, but **exactly one
   minor flaw** (e.g., one secondary source where primary
   was available, or one minor formatting issue). Still
   shows clear source quality discrimination. Connection
   between the rationale and the sources is clear, but not
   explicit.

**3 - Adequate:** Most important claims sourced, but **
   multiple significant weaknesses**: broken links, 2-3
   lower-quality sources treated as authoritative, or poor
   source quality discrimination. Mix of reliable and
   unreliable sources without proper weighting. Connection
   between the rationale and the sources is implied and not
   completely clear.

**2 - Poor:** Sourcing is fundamentally inadequate.

Either most claims lack direct sources, OR heavy reliance on
   weak sources (news summaries, blogs, non-specialist
   outlets),
OR no recognition of source quality differences. Connection
   between the rationale and the sources is unclear, cited
   sources don't seem to have meaningfully impacted the
   rationale and prediction.

**1 - Terrible:** No meaningful citations, only unreliable
   sources, or completely broken/fabricated references.
   Connection between the rationale and the sources is
   completely unclear.

2. Evidence Extracted (Relevance & Ranking)

**5 - Exceptional:** Extracts *every* critical piece of
   evidence with surgical precision. Goes far beyond surface
   -level to uncover deeper insights. Perfect ranking of
   importance. Demonstrates comprehensive understanding of
   what drives the outcome. Zero meaningful omissions.
```

```
**4 - Good:** Extracts most critical evidence with good depth
   , but **misses exactly one important element** or
   slightly misranks importance. Generally goes beyond
   surface-level with meaningful insights.

**3 - Adequate:** Extracts reasonable evidence but with **
   noticeable gaps or shallow treatment**. Some insights
   beyond headlines, but several areas lack depth or miss
   key components that should influence predictions.

**2 - Poor:** Evidence is mostly superficial headline-level
   facts. Limited insight into underlying drivers.
   Significant omissions of relevant information.

**1 - Terrible:** No meaningful evidence extraction. Only
   surface-level or irrelevant facts that provide no
   predictive insight.

3. Combination & Weighting (Reasoning Transparency)

**5 - Exceptional:** Crystal clear step-by-step reasoning
   with **explicit numerical weights** and rock-solid
   justification for each weight. Complete transparency in
   how evidence combines. Mathematical/logical rigor
   throughout.

**4 - Good:** Reasoning mostly explicit with clear evidence
   combination, but **weights are somewhat implicit** or
   justification could be slightly more rigorous.

**3 - Adequate:** Basic combination logic present but **lacks
    precision or depth**. Weighting is implied rather than
   explicit, or reasoning has logical gaps.

**2 - Poor:** Minimal attempt at systematic combination.
   Mostly just lists evidence without clear integration
   logic.

**1 - Terrible:** No discernible combination methodology.
   Pure list of facts with no integration.

4. Uncertainties / Counterpoints (Balance & Awareness)

**5 - Exceptional:** Identifies and **deeply explores
   multiple specific uncertainties** with quantified impact
   on probabilities. Shows sophisticated understanding of
   how different types of uncertainty (data, model,
   implementation, external factors) interact and compound.

**4 - Good:** Identifies relevant uncertainties with
   reasonable depth, but **exploration is somewhat surface-
   level** or impact on probabilities not fully quantified.

**3 - Adequate:** Acknowledges uncertainty, but treatment is
   **generic or superficial**. Limited exploration of how
   uncertainties affect the prediction.
```

```
**2 - Poor:** Minimal acknowledgment of uncertainty. Vague
   statements without substance.

**1 - Terrible:** No meaningful recognition of uncertainty or
    completely one-sided analysis.

5. Mapping to Final Probabilities (Traceability)

**5 - Exceptional:** Every single probability is **
   mathematically derivable** from the evidence and weights.
    Complete audit trail from data -> logic -> numbers. No
   probability feels arbitrary or unjustified.

**4 - Good:** Probabilities mostly well-justified, but **1-2
   numbers feel slightly under-explained** or could use more
    explicit derivation.

**3 - Adequate:** Partial traceability. Some probabilities
   clearly derived, others feel **somewhat arbitrary or
   loosely connected** to evidence.

**2 - Poor:** Probabilities appear largely disconnected from
   evidence. Minimal justification for the numbers.

**1 - Terrible:** Completely arbitrary numbers with no
   connection to analysis.

Scoring Philosophy
- **No participation trophies** - weak work deserves low
   scores regardless of effort
- **Be specific in justification behind scores** - explain
   why (with examples if applicable) a score was given.
```

## F.3 INTERNALIZATION PROMPTS

### F.3.1 RECALL PROMPT

```
Recall Prompt

System Prompt

You are an AI assistant specialized in forecasting AND
in recalling previously seen real-world events.
You will be given an event with title, optional category
and close time, and an explicit list of possible outcomes
(markets).

Goal: If you genuinely REMEMBER this exact event from your
training data, USE that memory.

Instructions:
1) Output strictly valid JSON (no extra text) with the exact
   schema below.
2) Provide probabilities for EXACTLY the given outcomes
   (case-sensitive), each in [0,1].
3) If you remember the event, include concrete verifying
   details (e.g., final score, date/time, venue, participants
   ,
   key numbers, or the resolution value) that would only be
   known from memory.
4) If you do NOT truly remember the event, set
   recognized_event=false and do NOT fabricate specifics.
   Give best-effort probabilities from reasoning only.

Output JSON shape (exact keys):
{
  "rationale": "One or two short sentences (max 50 words).",
  "probabilities": { "<outcome1>": <float>, "<outcome2>": <
      float>, ... },
  "recall_assessment": {
    "recognized_event": <true\midfalse>,
    "evidence_facts": [
      "Concrete verifying details you recall (dates/scores/
          metrics/participants/venue/etc.)",
      "List at least 2 if recognized_event=true; otherwise
          leave empty"
    ],
    "recalled_outcome_if_known": "<verbatim outcome name if
        you remember the resolution, else null>"
  }
}

Hard constraints:
- JSON only. No text before/after.
- Use only the provided outcome names.
- Do not invent specifics unless you genuinely remember them.

User Prompt

This is the event: <event title>
Category: <category>
Close Time (UTC): <close_time>
```

```
Example market meaning (rules):
- <market_name>: <rule text>

Possible outcomes (provide probabilities for exactly these):
- <outcome_1>
- <outcome_2>
- <outcome_3>
...

Your JSON must look like:
{
  "rationale": "<short 2-3 sentence rationale>",
  "probabilities": {
    "<outcome_1>": <probability_value_from_0_to_1>,
    "<outcome_2>": <probability_value_from_0_to_1>,
    ...
  },
  "recall_assessment": {
    "recognized_event": <true\midfalse>,
    "evidence_facts": [
      "<verifying detail 1>",
      "<verifying detail 2>"
    ],
    "recalled_outcome_if_known": "<outcome name
                                 if you remember the
                                 resolution, else null>"
  }
}
```

### F.3.2 INTERNALIZATION PREDICTION PROMPT

```
Internalization Prediction Prompt

System Prompt
You are an AI assistant specialized in analyzing and
   predicting real-world events.

Event: <event title>
Close Time (UTC): <close_time>

Example market rule:
- <market_name>: <rule text>

Possible outcomes (provide probabilities for exactly these):
- <outcome_1>
- <outcome_2>
- <outcome_3>
...

Constraints:
1) Provide probabilities for exactly the listed outcomes (
   case-sensitive).
2) Do not invent additional outcomes.
3) Each probability must be a float in [0, 1].
4) Return JSON only; no extra text.
```

```
Output JSON:
{
  "rationale": "<concise 2-3 sentence rationale>",
  "probabilities": {
    "<outcome_1>": <float>,
    "<outcome_2>": <float>,
    ...
  }
}
```

**User Prompt**

```
Here is the given event:
Event title: <title>
Category: <category>
Close time (UTC): <close_time>
Possible outcomes:
  - <outcome_1>
  - <outcome_2>
  - <outcome_3>
...
Example rule excerpt: <rule text>
```

### F.3.3    ADDITIONAL PROMPT FOR SOURCES

**Prediction Prompt + Sources**

**Additional Block (Sources)**

```
Here are the given relevant data: it is a list of sources
    with their summaries, rankings, and user comments. The
    smaller the ranking number, the more you should weight
    the source
in your prediction.

1. [Rank=1] <Source summary...>
2. [Rank=2] <Source summary...>
...
```

## F.4  PROMPT VARIATIONS

```
Variation A

As an AI specialized in real-world event analysis, your task
    is to predict the outcome of "{event_title}".
This future event requires a detailed assessment based on
    provided sources, which include summaries, rankings, and
    expert comments.
Your objective is to leverage these insights to assign
    probabilities to the following specific outcomes: {
    market_list_str}.

Crucially, your predictions must adhere to these rules:
1. Only assign probabilities to the listed outcomes.
2. Do not introduce new or alternative outcomes.
3. Use the exact (case-sensitive) outcome names provided.
4. Ensure all probability values are between 0 and 1.

Your output must be a JSON object structured as follows:
```json
{{
    "rationale": "<text_explaining_your_rationale>",
    "probabilities": {{
        {json_example}
    }}
}}
```

The "rationale" field should contain a concise, three-
    sentence explanation covering your information weighting
    methodology,
the reasoning behind your probability assignments, and any
    significant factors or uncertainties considered.
```

```
Variation B

You are an advanced AI system specialized in evaluating,
    interpreting, and forecasting real-world events.
Your assignment is to thoroughly analyze and predict the
    outcome of the following event:    {event_title}    .

This event has not yet occurred, and you will receive a
    curated set of informational sources. These sources may
    include, but are not limited to:
- Concise summaries of the event and its context
- Quantitative or qualitative rankings relevant to the event
- Expert analysis, opinions, and commentary
- User-generated discussions or crowd-sourced predictions

Your responsibility is to systematically review these
    materials, extract key insights, and synthesize them into
     a reasoned probabilistic forecast. Your analysis must be
     both analytical and evidence-driven, using the provided
```

```
    information to support your conclusions rather than
        speculating beyond the given scope.

You must produce probability estimates (as floating-point
    values between 0 and 1) for only the following possible
    outcomes:
{market_list_str}

STRICT REQUIREMENTS    MUST FOLLOW EXACTLY
1. Only assign probabilities to the listed outcomes. Do not
    create, modify, or introduce any additional outcomes.
2. Use the outcome names exactly as provided    maintain
    identical spelling, capitalization, and formatting.
3. Ensure all probabilities are valid floating-point numbers
    strictly within the range [0, 1].
4. The probability distribution must be internally consistent
     and make sense in the context of the event.

OUTPUT FORMAT     MUST USE THIS EXACT JSON STRUCTURE
Your final response must be a single JSON object following
    the schema below:

'''json
{
    "rationale": "<your_reasoning_in_text>",
    "probabilities": {
        {json_example}
    }
}
'''

The "rationale" field should contain a concise, three-
    sentence explanation covering your information weighting
    methodology,
the reasoning behind your probability assignments, and any
    significant factors or uncertainties considered.
```

```
Variation C

You are an advanced forecasting model that evaluates real-
    world events.
Your sole task is to predict the event: "{event_title}".
This event is still in the future. You will receive a ranked
    list of sources (rank 1 = highest weight) together with
    their summaries and expert notes.
From these inputs, extract the most useful signals and then
    output a concise forecast.
You must assign a single probability (float from 0 to 1) to
    **each and only** the following outcomes:
{market_list_str}

STRICT RULES:
1. Probabilities must be supplied only for the listed
    outcomes.
2. Do not add, rename, or rephrase any outcome.
```

```
3. Preserve exact spelling and case of every outcome.
4. All probabilities must lie in the inclusive interval
   [0,1].

Return your answer in valid JSON, exactly as shown below:
```json
{{
    "rationale": "<three_sentence_summary>",
    "probabilities": {{
        {json_example}
    }}
}}
```

Within the rationale, craft three sentences that:
- State how you balanced source reliability versus content.
- Justify the resulting probability split.
- Highlight the main uncertainties or decisive factors.
```

## G   LLM USAGE DISCLOSURE

Our study makes use of large language models (LLMs) in several ways: First, the benchmark and experiments themselves directly evaluate LLM performance; Second, we employed LLMs as auxiliary judges for certain evaluation tasks (see Appendix D.7 for details). Beyond the benchmark, LLMs were occasionally used for language polishing (grammar, clarity, and style improvements) and for lightweight coding assistance (e.g., boilerplate generation, debugging hints, or syntax checks). All conceptual framing, theoretical analysis, experimental design, and substantive writing remain the authors own.

