# OpenReview forum: "LLM-as-a-Prophet: Understanding Predictive Intelligence with Prophet Arena"
_ICLR.cc/2026/Conference — ICLR 2026 Poster_

### Official Review · Reviewer_Uttz · 2025-10-29

**Soundness:** 2
**Presentation:** 3
**Contribution:** 1
**Rating:** 2
**Confidence:** 5

**Summary:**

The paper aims to explore whether large language models (LLMs) possess genuine predictive intelligence. The authors argue that forecasting can serve as a rigorous and unified test of intelligence, as it integrates reasoning, calibration, and evidence aggregation.

**Strengths:**

1) The paper is overall well written and easy to follow.

2) The study is methodologically sound and supported by a comprehensive empirical evaluation. However, the benchmark’s reproducibility and transparency are limited, as key implementation details (event selection, context construction) are not fully released.

3) The work offers a valuable new perspective on evaluating LLMs’ predictive and reasoning abilities. While the conceptual framing is interesting, the paper’s technical contribution remains limited. More importantly, a benchmark’s ultimate value lies in its ability to guide future model development by revealing actionable weaknesses or providing diagnostic insights. Prophet Arena, however, stops at evaluation and does not clearly demonstrate how its results can inform the improvement of model architectures, reasoning mechanisms, or calibration strategies. The work could be strengthened by explicitly linking benchmark findings to concrete modeling directions or by releasing more reproducible implementation details.

**Weaknesses:**

1) Although the paper claims to release part of the dataset, the full event selection, context construction, and preprocessing pipeline are not disclosed. No implementation details, scripts, or reproducible framework are provided for replicating Prophet Arena. As a result, the benchmark cannot be independently reproduced or audited, which weakens its scientific transparency.

2) Prophet Arena primarily repackages existing components—prediction market data, web retrieval, and standard evaluation metrics (Brier score, ECE, return). The framework does not introduce new modeling techniques or algorithmic insights, serving mainly as an evaluation setting. Comparisons are limited to a single market-based baseline, lacking systematic benchmarks against structured forecasting models or classical probabilistic methods.

3) The paper proposes using “real-world forecasting” as a proxy for intelligence, but does not justify why this task is representative, stable, or objectively measurable. It remains ambiguous whether Prophet Arena is intended as a benchmark, a task suite, or a conceptual lens for studying intelligence.

4) (Important) The paper does not explain which cognitive abilities are actually being measured (e.g., reasoning, retrieval, knowledge recall, uncertainty estimation). Without disentangling these capabilities, the source of success or failure in forecasting is unclear. This makes the benchmark weak in diagnostic interpretability and limits its use for targeted model improvement.

5)  Experimental results show that LLMs do not outperform the market baseline in prediction accuracy or expected return. The models’ predictions are often overly conservative or poorly calibrated, as acknowledged by the authors. Hence, the study fails to demonstrate real-world utility or unique advantages of LLMs in open-domain forecasting.

6) (Minor Points) Figure 10 is blurry, and Figure 11 (right) has a formatting issue — the “Category” label is not fully visible.

**Questions:**

1) Do the authors view Prophet Arena primarily as a diagnostic tool for understanding LLM reasoning, or as a performance benchmark for future model comparison?

2) How can the results from Prophet Arena inform or guide the next stage of model development? For example, can specific reasoning or calibration failures observed in forecasting tasks translate into actionable model improvements?

3) Since the event selection, context construction, and preprocessing pipelines are not fully released, how do the authors ensure reproducibility and fairness across different models and time periods?

4) Other questions are discussed in the Weakness section.

---

> ### Author Response · Authors · 2025-11-23
> **Response to reviewer Uttz**
>
> We appreciate the reviewer's detailed feedback, though we respectfully point out that some of the points raised may be due to major misperceptions of our paper. We hope our clarifications below could help the reviewer re-evaluate the merit of our work.
>
> ### **W1 Limited Reproducibility: The event selection and context pipeline are not disclosed.**
>
> We respectfully disagree with this premise as our implementation details are documented extensively in **Appendix A**, and all prompts used throughout the pipeline are fully disclosed in **Appendix E**. In addition, we provide a dataset on Hugging Face. We are also happy to inform the reviewer that since the initial submission, we have released a substantially larger dataset (1,000 event submissions, roughly 10× the size of the original release) on hugging face as well.
>
>
> We also wish to highlight that our evaluation setting differs from that of other standard benchmarks, imposing a different set of constraints of what or what cannot be shared public. Specifically, opting to make the event-sourcing script public would enable models to directly recognize or retrieve the exact evaluation events, potentially leaking information and compromising the validity of the benchmark. For this reason, we make all event data and evaluation code available, while withholding only the specific sourcing script that would allow direct reconstruction of the held-out test set.
>
>
> ###  **W2 Limited technical contribution**
>
> We would like to clarify **that our contribution is an evaluation framework rather than a new model or algorithm**. To our knowledge, this is the first work to unify open-domain forecasting with a comprehensive set of metrics that reflect practical forecasting performance: prediction accuracy, calibration error (ECE), and expected financial return. This metric suite is important for capturing different dimensions of real-world forecasting quality.
>
> Moreover, Prophet Arena is intentionally modular. This design enables researchers to isolate and study specific components of the forecasting pipeline, such as retrieval quality, reasoning behaviors, or probability calibration. Finally, by incorporating a market consensus, our benchmark provides a practical reference for assessing state-of-the-art LLM forecasts, as no other structured forecasting models are well-suited for this open-domain task. Even if such models exist, their predictions are likely already reflected in the consensus of highly liquid prediction markets.
>
>
>
>
> ### **W3 & Q1 On the purpose of our work**
>
> Prophet Arena is first and foremost **a benchmark for LLMs on real-world forecasting**. However, our modularized design and evaluation metrics are deliberately constructed so that Prophet Arena can simultaneously serve as a task suite and provide a rigorous lens on predictive reasoning.
>
> We agree that “intelligence” is a broad construct, and our goal is not to claim that real-world forecasting fully captures all aspects of intelligence. Rather, we focus on one aspect: the ability to form accurate, calibrated probabilistic beliefs about unresolved real-world questions.
> Human forecasting has already been standardized around this notion: large-scale tournaments (e.g., Good Judgment, M-competitions) and the recent surge of interest in prediction markets (e.g., Kalshi, Polymarket) evaluates forecasters on streams of diverse, real-world questions through multiple evaluation metrics. This precisely addresses the three properties the reviewer highlights:
> - Representative: question sets span geopolitics, economics, science, and technology, and performance on these events is predictive of broader reasoning skills and downstream decision quality.
> - Stable: scores aggregate over a large body of questions and are used in practice for ranking, training, and selecting human forecasters over long periods.
> - Objectively measurable: outcomes are externally resolved and scored by proper scoring rules, yielding comparable, incentive-compatible metrics.
>
> Prophet Arena extends this existing human standard to LLMs: we take the same style of externally resolvable, real-world questions and evaluate models with rigorous scoring rules. In our paper, we use real-world forecasting as an operational, objectively measurable proxy for predictive intelligence.

---

> ### Author Response · Authors · 2025-11-23
> **Response to reviewer Uttz**
>
> ### **W4 & Q2: On the diagnostic interpretability of our benchmark**
>
> We respectfully disagree with the reviewer’s claim that the benchmark lacks diagnostic interpretability. The core of our contribution is an explicit modular decomposition of the forecasting pipeline, which is designed precisely to enable capability-level analysis. **Section 4** systematically identifies bottlenecks in current LLMs across reasoning (**Sec. 4.3), retrieval (**Sec. 4.1.1**), and knowledge recall (**Sec. 4.1.2**) – the very cognitive abilities the reviewer posits are unmeasured and not included.
>
> The comment on “uncertainty estimation” similarly overlooks a core aspect of the benchmark. Models must output probabilistic forecasts, which are their uncertainty estimates, and we evaluate these directly using ECE-based calibration metrics. Appendix D.3 further examines how different probability-verbalization strategies affect calibration and forecast quality. These analyses directly probe models’ uncertainty handling.
>
> While we acknowledge that fully disentangling forecasting-relevant capabilities remains an open research challenge, Prophet Arena already offers a structured and extensible platform for systematically measuring these dimensions. Its modular design is specifically intended to support targeted diagnostics and to inform future model development.
>
>
> ### **W5: LLMs don't beat the market, which is a weakness of the paper.**
>
> We would like to respectfully remind the reviewer that our results do not support the claim that “LLMs do not outperform the market baseline.” As shown in **Table 2**, the frontier proprietary models consistently outperform the Market Baseline across all metrics. While our paper shows several aspects and scenarios where LLM performance is lacking, it does not invalidate the promise of “LLM-as-a-prophet”.
>
> We also want to highlight that **beating the market is a billion-dollar challenge, not a prerequisite for developing a benchmark nor what our contribution sets out to do**. The purpose of Prophet Arena is to set a rigorous baseline using the strongest human consensus (the market). The fact that LLMs fail to meet this high bar is a critical scientific finding, not a flaw in our evaluation design. We view the shortcomings of frontier models rather as a signal that this is a fertile domain for more future work and progress.
>
>
> ### **Q3: how do the authors ensure reproducibility and fairness across different models and time periods?**
>
> All models are evaluated on the same set of events given the exact same prediction context as the input. We refer the reviewer to our implementation details in **Appendix A** as well as the prompts we used in **Appendix E** for further details.
>
>
> Lastly, we acknowledge the suggestions to improve the resolution of the figures and have updated the manuscript accordingly. Thank you!

---

> > ### Comment · Reviewer_Uttz · 2025-11-26
> >
> > Thank you for the authors’ response, which has addressed several of my concerns. I am therefore raising my score to 4, and I would be open to further increasing it if the points below can be clarified in more detail. However, I still have the following reservations:
> > (1) The assumption that LLMs can forecast future events lacks theoretical grounding and currently relies mostly on empirical observations. Even if this premise is accepted, it remains unclear what practical guidance it offers—for instance, for model selection or model training.
> > (2) It remains unclear whether some of the benchmarked tasks are genuinely predictable. For example, prediction markets contain noise and non-rational behaviors, so their suitability as an evaluation baseline requires more thorough justification.
> > (3) Although withholding the sourcing script helps prevent data leakage, it nevertheless affects the reproducibility and transparency of the benchmark.

---

> > > ### Author Response · Authors · 2025-11-27
> > >
> > > We sincerely appreciate the reviewer’s engagement, thoughtful feedback, and openness to further increasing the rating. We believe our clarifications below should help fully address the reviewers’ points, and are happy to hear any further comments if the reviewer might have.
> > >
> > > ## (1) On LLMs as forecasters and practical guidance the benchmark offers
> > >
> > > It is true that LLMs’ capability in forecasting future events has demonstrated strong empirical performance, even though there remains a lack of theoretical grounding. However, we respectfully point out that this is a shared phenomenon across most applications of LLMs, including Gold-medal-winning LLMs in math and informatics Olympiad, expert-level chatbots such as ChatGPT, and expert-level coding agents such as Claude Code, all of which currently lacks theoretical grounding. While developing rigorous theory for these capabilities, including forecasting, is an important and fascinating research direction, the present lack of theory should not be interpreted as undermining the contribution of our work, just as it has not diminished the scientific value of other widely recognized LLM advances.
> > >
> > > Regarding how our benchmark guides model training and selection, two types of models need to be considered separately. For **general-purpose LLMs**, such as Gemini 3 and GPT 5, their training and selection **should, by definition, not be guided by their performances on a specific benchmark** since these models are trained to acquire *general intelligence*, whereas almost all benchmarks are testing specific intelligence such as reasoning (e.g., BIG-Bench), alignment (e.g., LMArena) and coding (SWE-Bench). For **specialized models** targeting a concrete application domain (forecasting in our case), **our benchmark does offer guidance on how to train and select models**. In fact, the entirety of Section 4 offers diagnostic analyses about current strengths and limitations of LLMs in forecasting, offering pathways for future training and improvement of models specialized in forecasting.
> > >
> > >
> > > ## (2) On predictability of tasks and the use of prediction markets
> > >
> > > We appreciate the insightful question on the predictability of some tasks, and address this question from a few angles. First, **our empirical results already demonstrate that these tasks are mostly predictable**, in the sense that the forecasts of all models as well as the market are strictly better than a random guess. This is evident from models’ Brier scores as shown in Table 2, which are all better/smaller than the Brier score of random guess (i.e., 0.25). Second, **the existence of a few difficult-to-predict tasks do not affect our evaluation**, since our evaluations are averaged over a large number(>500) forecasting tasks.
> > >
> > > Regarding the reviewer’s concern about the stability of prediction markets as a baseline, we believe this could be fully resolved from a few angles. First, the efficacy of prediction markets as a forecasting tool is deeply grounded in extensive economic research. Numerous past papers have emphasized this point. In particular, a representative paper is the seminal work titled [*The Promise of Prediction Market*](https://www.science.org/doi/10.1126/science.1157679) at **Science 2008**, authored by a celebrated list of economists (including 5 Nobel Laureates). The key economic insight is that prediction markets efficiently incentivize participants to collect and aggregate their information, facilitating the aggregation of collective wisdom. Second, while there indeed exists irrational and noisy actors, most participants tend to be rational due to strong incentives involved. Hence, the aggregated outcome shown on the market tends to be accurate. To provide the reviewer some concrete numbers, **Polymarket’s [recent data](https://polymarket.com/accuracy) shows that they have 95.1% accuracy 4 hours before event resolution and 91.4% accuracy 1 month before event resolution.**
> > >
> > >
> > >
> > > ## (3) On open sourcing our event sourcing scripts
> > >
> > > As stated earlier, releasing the exact sourcing script would enable models (or users) to pre-collect **the identical** set of future contexts used on our platform, comprising the purpose of our live and contamination-resistant benchmark. However, we fully agree that transparency is essential for a benchmark and reproducibility is critical for research findings.
> > >
> > > To address this balance, the Appendix provides detailed implementation information for our evaluation pipeline, including prompts (**Appendix F**), sourcing techniques (**Appendix B**), and more. In addition, to further clarify and support future development, we now release extensive technical documentation along with an example Kalshi event sourcing script in the **supplemental materials.**
> > >
> > > We thank you again for the thoughtful discussion!

---

> > > > ### Comment · Reviewer_Uttz · 2025-11-27
> > > >
> > > > Thank you for the authors’ efforts. I have re-evaluated the contribution of this work and am inclined to raise my score to 6 (positive). My only remaining question is whether there is a clear update and maintenance plan to ensure the benchmark remains contamination-free as it evolves.

---

> > > > > ### Author Response · Authors · 2025-11-27
> > > > >
> > > > > We appreciate the reviewer’s positive assessment of our work and thoughtful engagement. The answer to the reviewer’s question is absolutely a **Yes** – in fact, our team has been constantly maintaining (even expanding) the Prophet Arena platform after the ICLR submission and continuously collecting new events every day to ensure that all models continue to be evaluated on unresolved events, hence ensuring the evaluation to be perpetually contamination-free.  Below, we take this opportunity to further clarify our plans on keeping Prophet Arena contamination-free as it grows.
> > > > >
> > > > > Our approach will follow the design described in  **Section 2** and  **Appendix B**. Prophet Arena, regardless of future changes, will always be live and contamination-free by construction: we only evaluate models on **unresolved** prediction-market events at the time the model is queried and will continue to adhere to this design.
> > > > >
> > > > > As for our public releases of datasets, we will continue to have all events precisely time-stamped across market creation, context snapshot, and resolution, not only allowing for analysis but careful filtering by others who may use this data. This ensures that future models can be evaluated on contamination-free subsets even as the benchmark expands.

---

### Official Review · Reviewer_Wxhc · 2025-10-29

**Soundness:** 4
**Presentation:** 4
**Contribution:** 3
**Rating:** 6
**Confidence:** 3

**Summary:**

This paper proposes a live benchmark for evaluating LLMs’ ability to forecast future real-world events over time, addressing a core question with direct relevance to industrial decision-making. The results demonstrate both promising predictive intelligence and persistent bottlenecks in reasoning, evidence integration, and temporal information assimilation.

**Strengths:**

- This paper proposes a live and realistic benchmark that directly measures LLMs’ forecasting ability on future real-world events, highly relevant to industrial applications.

- Strong empirical design including probabilistic scoring, calibration, and market-based evaluation, providing a multi-angle understanding of model forecasting capability.

- Clear analysis connecting reasoning quality and belief update dynamics.

- Well-articulated motivation and clean methodology presentation.

**Weaknesses:**

- No measurement of alignment between human judges and model judges, raising concerns about rating validity.
- Reliance on a single web-search agent without explicit evidence quality assessment may confound reasoning evaluation.

**Questions:**

1. Can the authors report performance differences across prediction horizons (short vs mid-term)? Since most events resolve within days or hours, current results may reflect late-stage market imitation rather than true foresight. Distinguishing performance across different lead times would reveal whether models are actually anticipating future outcomes, or simply reacting better as more information becomes available near resolution.

2. Retrieval module lacks noise-robustness. News access relies solely on a search agent (GPT-4o) with minimal noise filtering or source-quality controls. How to improve evidence quality?

3. Since LLMs function as judges without calibration against human scoring, there is a risk of alignment bias or shared model failure modes. Do the authors have plans for human adjudication or inter-rater agreement studies to validate reasoning evaluation?

---

> ### Author Response · Authors · 2025-11-22
> **Response to reviewer Wxhc**
>
> We thank the reviewer for their positive feedback and we address your questions and concerns below.
>
> ## **W1 & Q3: On the alignment between human judges and model judges**
> Thank you for raising these points. To address concerns regarding alignment, during the rebuttal period, we conducted a blind human adjudication study on 170 sampled events. Our detailed results can be found in **Global Response #2**. In summary, we found that human ratings closely match with ratings of the LLM-as-a-judge (mean absolute differences <0.5 on a 1–5 rating scale, with >94% within 1 point difference).
>
> ## **W2 & Q2: On the analysis of web-search agents and their retrieval quality**
> Excellent point, retrieval quality is indeed critical. In **Global response #1**, we expand on this domain, updating our retrieval module to have four searchers. While this exploration explains our selection of gpt4o as our primary searcher, we also want to emphasize that our forward looking plan is to continue evaluating new and improved searchers as the space continues to evolve.
>
> ## **Q1: Can the authors report performance differences across prediction horizons (short vs mid-term)?**
> In **Section 4.2** and **Figure 6**, we analyzed the performance of various models over different horizons, binning forecasts by time to resolution. Since ICLR submission, we have been continuously collecting forecasting data over more events. In the updated PDF, we have now additionally updated **Figure 6** of **Section 4.2** to reflect results that incorporating these recently collected data and also made it semantically clearer, to incorporate recent data collected after submission, and to include more models for comparison.
> The results from these experiments demonstrate a few interesting trends with regard to the short vs mid term performance. In the short‑term regime (last few hours before resolution), all models improve, but the market baseline improves more sharply than any model, leaving a noticeable gap between models and markets. This is somewhat expected since when it is close to resolution, markets with strong incentive structures are much more efficient aggregators of information, compared to LLMs.  At the mid-term regime (1-4 days before resolution), relative model performance is mostly uniform, with a few outliers improving or worsening performance, while absolute model performance slowly increases as resolution approaches. Finally, models show clearly advantages over markets at long-term regime (> 4 days). This longitudinal analysis of the performance of various models is now also updated in section 4.2 of the updated PDF.

---

> > ### Comment · Reviewer_Wxhc · 2025-11-26
> >
> > Thank you for adding more results and analyses on web search and the alignment between human and model judges. I also understand the challenges involved in predicting medium- and long-term events. My main concerns have been addressed, and I will maintain the positive scores.

---

> ### Author Response · Authors · 2025-11-27
>
> Thank you for your thoughtful review and constructive feedback. We’re glad we were able to address your concerns, and we appreciate the positive evaluation across the subscores and overall rating. If there are any remaining points you’d like us to clarify to *potentially warrant an increase in score* before the rebuttal window closes, we would be happy to resolve them.

---

### Official Review · Reviewer_kx9H · 2025-11-01

**Soundness:** 3
**Presentation:** 3
**Contribution:** 3
**Rating:** 6
**Confidence:** 3

**Summary:**

1. The manuscript perfectly proposes = LLM-as-a-Prophet, testing LLMs’ ability. This actually helps forecast future real world events
2. Introduces Prophet Arena, this is a live benchmark which is using prediction-market data and real-time web context.
3. The manuscript also evaluates array of 22 LLMs. This represents diverse number of large language of models.
4. The manuscript Measures forecasting loss (Brier), calibration error (ECE), and market return.
5. GPT-5R shows best calibration and lowest loss; all models still lag human markets.

**Strengths:**

1. The manuscript presents novel paradigm: which is real-world, contamination-free forecasting benchmark.
2. The manuscript also presents comprehensive metrics: which is a good blend of accuracy and calibration.
3. Good part is about the open-sourced subset for reproducibility.

**Weaknesses:**

1. The Event recall errors and approximate temporal memory which could be a potential weakness.
2. The weakness Conservative probability estimates vs markets.
3. Dependence on search/source quality; not all domains benefit equally.
4. Limited profitability (returns < 1).
5. Incomplete foresight near event resolution; calibration still imperfect.

**Questions:**

1. Can you tell me how robust are results to prompt or search-engine variations along with some metrics to compare and evaluate?
2. Could multi-agent or ensemble LLMs outperform single models and how this multi agentic workflow will be working together?
3. How to improve knowledge precision and can you tell me what can be done to improve it?
4. Can MCPs be used to do the same evaluation.

---

> ### Author Response · Authors · 2025-11-22
> **Response to reviewer kx9H**
>
> We sincerely thank the reviewer for the thoughtful assessment and constructive questions. We are glad that you found the contribution of our benchmark study – including its design, evaluation metrics and reproducibility – to be sound and well-motivated. We believe our rebuttal should help thoroughly address all of the weakness and questions from the reviewer, as we now elaborate below.
>
> ## **Clarifications on “Weaknesses”**
>
> We appreciate the concerns you raised; however, we would like to gently clarify that these raised points reflect limitations of current LLMs rather than weaknesses of our paper or methodology. Indeed, uncovering these bottlenecks is a central goal of Prophet Arena: as a **benchmark research** in nature, our platform is designed not to propose solutions, but to reveal where today’s frontier models fail in real-world, contamination-free forecasting (so as to guide future model and agent-system development). With this goal in mind, **we view the reviewer’s recognition of these LLM weaknesses, as unveiled by our work, as a success of our research, rather than a limitation**.
>
> ## **Q1: Robustness to prompt or search-engine variations**
>
> We also agree that robustness to prompts is important, and would like to gently point out that we have already conducted thorough robustness analysis. Specifically, we summarized our findings in **Section 4.1.4**, which links to **Appendix D.3** that contains details about our experiment design, implementation and in-depth analysis. In summary, through systematically varying prompt phrasing, we show that the relative ranking of models remains stable, offering evidence for the robustness of our analysis.
>
> In addition, Search-engine variation and its impact on forecasting quality is discussed thoroughly in our **Global Response #1**. In short, different searchers do change the model rankings slightly. However, top models perform consistently well in general across all searchers.
>
> ## **Q2. Could multi-agent or ensemble LLMs outperform single models?**
>
> This is an excellent and forward-looking question. While the present work does not evaluate multi-agent pipelines (which is also the case for majority of LLM benchmark works such as LMArena for benchmarking alignment [1] and BIG-Bench research for benchmarking reasoning capabilities [2]), we agree they could plausibly outperform single models, especially by combining various search strategies, diverse reasoning styles, and internal debate or aggregation mechanisms. We consider this an exciting direction for future work, and Prophet Arena is expressly built to support such experimentation.
>
> ## **Q3: How to improve knowledge precision?**
>
> We analyze this issue in detail in **Section 4.1.1**, where we show that many LLM errors arise from approximate temporal recall and imprecise extraction of real-time facts. Improving model grounding (via better retrieval, more structured search, and stronger temporal reasoning) is a key research challenge that our benchmark helps expose.
>
> ## **Q4: Can MCPs (or general tool-use agents) be used for evaluation?**
>
> Thanks for the insightful comment. Indeed, incorporating MCPs or other tool-augmented agents is an interesting future direction and could provide an evaluation framework for **agentic** forecasting pipelines. This, however, is not the focus of the present work, which aims to understand the language model’s raw capability in converting event-relevant contexts into event forecasts.  With that said, we agree that what the reviewer suggested is a compelling avenue for another research endeavor.
>
>
> [1] Chiang, Wei-Lin, et al. "Chatbot arena: An open platform for evaluating llms by human preference." Forty-first International Conference on Machine Learning. ICML 2024.
>
> [2] Suzgun, Mirac and Scales, Nathan and Sch{\"a}rli, Nathanael and Gehrmann, Sebastian and Tay, Yi and Chung, Hyung Won and Chowdhery, Aakanksha and Le, Quoc and Chi, Ed and Zhou, Denny and others, Challenging big-bench tasks and whether chain-of-thought can solve them, Findings of the Association for Computational Linguistics: ACL 2023.

---

### Official Review · Reviewer_reZn · 2025-11-01

**Soundness:** 3
**Presentation:** 3
**Contribution:** 3
**Rating:** 6
**Confidence:** 4

**Summary:**

The paper proposes evaluating LLM capabilities by forecasting live Kalshi questions. They setup an automated pipeline to extract, test, and resolve LLM forecasts. The evaluation results are summarized using both standard forecasting and calibration metrics like brier score and ECE, as well as (hypothetical) market returns and relative advantage between different LLMs. The paper has interesting analysis of LLM behaviours, ranging from: analysis across topics, effect of market data vs retrieved GPT 4o search results, different tendencies to assign extreme probabilities to forecasts, and analysis of reasoning traces and failure modes.

**Strengths:**

1. The paper introduces a useful live benchmark for measuring LLM capabilities via forecasting open Kalshi events.

2. I like the relative advantage based metrics used for comparing language models. This mitigates issues in existing benchmarks where questions can vary in difficulty (sometimes not even being "forecasting" questions as future information is leaked).

3. The paper has detailed analysis of LLM forecasting behaviour and interesting insights across diverse ablations in Section 4 and the appendix.

**Weaknesses:**

1. The paper does not show awareness of existing literature in LLM forecasting. For example, probabilistic forecasting is mentioned as the first "distinguishing feature" of the benchmark in the introduction. However, this has been the standard used in existing papers in LLM forecasting [1]. I am also concerned about the supposed "introduction" of "LLM-as-a-prophet paradigm". LLM forecasting has been an active area of study for the last 3 years [2]. I do not see the value of adding a new term, especially given the connotations of the word "prophet".

2. The use of GPT-4o based search as context across models could be a confounder. It is perhaps possible that the OpenAI models perform better on this benchmark because the retrieved context is more "in distribution" for them (as it comes from GPT 4o search + outputs) than other model families. Further, GPT 4o search is more likely to surface external context that bridges knowledge gaps in OpenAI models.

[1] Approaching Human-Level Forecasting with Language Models. Danny Halawi, Fred Zhang, Chen Yueh-Han, Jacob Steinhardt

[2] Forecasting Future World Events with Neural Networks.
Andy Zou, Tristan Xiao, Ryan Jia, Joe Kwon, Mantas Mazeika, Richard Li, Dawn Song, Jacob Steinhardt, Owain Evans, Dan Hendrycks

**Questions:**

1. I am slightly confused about the "hypothesis" / framing in the Introduction about predicting the next word leading to predicting the next event. Its definitely an interesting hypothesis, but one that can only be tested with base models without post-training which moves models away from "next-token-prediction" behavior. Could you either shift this hypothesis to be less prominent, or test it?

2. It would be useful to shift the benchmark construction methodology to the main paper, as currently the main paper barely talks about it. To make space, section 3.1, 3.2, and 4.1.1 can be shifted to the appendix as they are not unique to this work.

---

> ### Author Response · Authors · 2025-11-22
> **Response to reviewer reZn**
>
> We thank the reviewer for the insightful comments and for the positive assessment of the benchmark’s soundness, presentation, and relevance. During the rebuttal period, we are able to thoroughly address all of the reviewer’s comments on the weakness of our work via additional research, as elaborated below.
>
> ## **W1: On lack of discussions of related work**
>
> Due to space limitations, we were unable to include a more detailed discussion of prior work in the initial submission. Following your suggestion, we have now highlighted the most relevant related studies in the main text (see **Line 80-110** of the updated PDF), as well as acknowledging earlier works on the design of AI forecasters in **Section 2**. Although existing papers touch on elements of AI forecasting, we note that this area remains comparatively overlooked, especially relative to the community’s current focus on mathematical reasoning and other structured tasks. Hence, a major goal of our paper is to highlight forecasting as a core capability of general intelligence and to motivate the concept of LLM-as-a-Prophet through the analysis of data collected from our benchmark.
>
> ## **W2: On reliance on a single web-search agent**
> See our **Global Response #1**. In short, we conducted extensive experiments with human raters (within our capability during the rebuttal period) for around 170 randomly sampled forecasting questions. Our evaluation shows small deviation between human raters and LLM-as-a-judge, offering evidence of reliability for the use of LLM-as-a-judge for our setting.
>
>
> ## **Q1: The hypothesis of “next token prediction to future event prediction”**
> We thank the reviewer for pointing out this insightful comment. Our original intention was *not* to propose this as a “hypothesis” in formal scientific sense, but rather viewing it as an exciting **prospect** of LLMs. In response, we have now changed the word “hypothesis”  on **Line 43** (which only shows up once) to “prospect” to make it less prominent. With that said, we agree with the reviewer that the formulation of a formal scientific hypothesis that *next-token prediction leads to future event prediction* is intriguing – though out of the scope of our current paper – and we expect the answer might be quite open-ended.
>
>
> ## **Q2: Suggestions for restructuring the paper writing**
> We sincerely appreciate this writing suggestion of reducing the metrics discussions and including details of the benchmark construction, which we deeply contemplated while drafting our paper. Ultimately, we felt that pointing out the value of using metrics like Brier score and ECE, as opposed to 0-1 loss, might be more valuable to the community as most previous works overlooked the choice of metrics for forecasting. For instance, all the following recent papers [1,2,3,4,5] solely use 0-1 loss to measure forecasters’ performance, which is something the standard forecasting literature rarely does due to limited expressiveness of accuracy.
>
> With that said, the reviewer’s comment has prompted us to seriously reconsider the suggestion. In the current updated version, we have not yet implemented the proposed restructuring, as doing so would require substantial reorganization of the paper – primarily due to space constraints rather than missing content, since all relevant material is already included. Meanwhile, if the reviewer has any further thoughts, we would be more than happy to hear during the rebuttal period.
>
>
> [1] Zhihan Zhang, Yixin Cao, Chenchen Ye, Yunshan Ma, Lizi Liao, and Tat Seng Chua. Analyzing temporal complex events with large language models? a benchmark towards temporal, long context understanding. In Long Papers, pp. 1588–1606. Association for Computational Linguistics (ACL), 2024
>
> [2] Hui Dai, Ryan Teehan, and Mengye Ren. Are llms prescient? a continuous evaluation using daily news as the oracle. In Forty-second International Conference on Machine Learning (ICML), 2025.
>
> [3] Zhen Wang, Xi Zhou, Yating Yang, Bo Ma, Lei Wang, Rui Dong, and Azmat Anwar. Openforecast: A large-scale open-ended event forecasting dataset. In Proceedings of the 31st International Conference on Computational Linguistics, pp. 5273–5294, 2025.
>
> [4] Jack Wildman, Nikos I Bosse, Daniel Hnyk, Peter Muhlbacher, Finn Hambly, Jon Evans, Dan ¨ Schwarz, Lawrence Phillips, et al. Bench to the future: A pastcasting benchmark for forecasting agents. arXiv preprint arXiv:2506.21558, 2025.
>
> [5] Zhiyuan Zeng, Jiashuo Liu, Siyuan Chen, Tianci He, Yali Liao, Jinpeng Wang, Zaiyuan Wang, Yang Yang, Lingyue Yin, Mingren Yin, et al. Futurex: An advanced live benchmark for llm agents in future prediction. arXiv preprint arXiv:2508.11987, 2025.

---

> > ### Comment · Reviewer_reZn · 2025-11-22
> >
> > Thanks for the new result with different searchers. I agree that the OpenAI models do seem strong across searchers. One thing I was a bit surprised by was the high variance in rankings based on the search model used. Do you have any intuition for why this happens? This is a little concerning if one searcher is used going forward.
> >
> > Thanks for adding a related work section. I disagree with the characterization "In contrast, the aforementioned
> > benchmarks largely emphasize dataset construction and only evaluates forecasting accuracy...". Note that Zou et al (2022), Halawi et al (2024) both contributed new test sets for forecasting, based on prediction markets (similar to your paper), and measured brier score. I really do not think measuring brier score / ECE is a contribution of this work, and would prefer it not be positioned this way.  Moreover, upon another reading, I am confused by the motivation to use ECE in L223-224. The brier score is also a proper scoring rule, and thus "trustworthy" according to your definition of that term. In what situations would one refer to ECE instead of brier score when comparing models? Could you please make this distinction clearer for readers?
> >
> > I continue to recommend shifting the benchmark construction methodology to the main paper. For anyone working in this domain, the primary question they would have before considering using your benchmark will be understanding how it was created. In many ways, that is the main contribution of this work, so not having at least the main methodology (details can stay in the appendix) in the main paper significantly hurts the presentation.
> >
> > I would also really like to know what is the topic-wise composition of the dataset used for the evaluations reported in the paper. From my understanding, Kalshi is turning into mostly a sports gambling and US politics platform. It is important for readers to know what knowledge/skill models are really being tested on under the hood in this benchmark.
> >
> > I am open to increasing my score to 8 if the above concerns are addressed. I may reduce my score to 4 if the topic-wise composition of the dataset and benchmark construction methodology are not clarified in the main paper.

---

> > > ### Author Response · Authors · 2025-11-25
> > > **Followup with reviewer reZn**
> > >
> > > We thank the reviewer for actively engaging with our updates.  We have taken the reviewer’s suggestions seriously, and believe that the updated PDF draft should have now fully addressed the reviewer’s suggestions/concerns. Details are also elaborated below, and any further comments are welcomed.
> > >
> > > ## **Variance in predictor rankings and enabling searchers going forward**
> > >
> > > We appreciate your observation about the high variance in rankings across searchers. We would like to note that **there are multiple active searchers (including grok-4-fast and gemini-2.5-pro) enabled now and going forward**.
> > >
> > > To explain the variance in rankings across searchers, our setup is a two-stage pipeline: (1) a searcher maps the question to a set of collected sources, and (2) a predictor maps those collected sources to probabilities. Intuitively, changing the searcher therefore changes the distribution of evidence each predictor sees (coverage, recency, stance, noise, etc.), so the “effective task” is slightly different under each searcher. Predictors differ in how robust they are with respect to these shifts, which shows up as variation in rankings when we condition on a single searcher’s outputs.
> > >
> > > As such, we view the choice of searcher as part of the benchmark configuration. For the purposes of this paper, fixing a single reliable searcher in the main results (gpt-4o) defines a clear, interpretable setting and already reveals substantial, systematic differences between predictors. The multi-searcher analysis we provide here should, therefore, be seen as an additional robustness check and diagnostic, rather than a requirement for every use of Prophet Arena, and we believe that establishing and carefully analyzing this single-searcher configuration already presents useful and substantial insights. Going forward, we will continue to collect data on multiple searchers for more in-depth analysis.
> > >
> > > ## **ECE vs. Brier score, and Revision on the  Metrics Section**
> > > First, we took the reviewer’s suggestion, and have combined the original Section 3.1, 3.2, 3.3 now to a single **Section 3.1**, **significantly scaling down the contents of metrics descriptions** (deferring all technical descriptions now to Appendix) and **toning down contribution descriptions** (e.g., referring to earlier works that have considered Brier score or calibration in the first paragraph of **Section 3**).
> > >
> > > Accompanying the reduced metrics descriptions, we enriched the discussions about the differences among these metrics. Particularly, we added a few lines at the end of **Section 3.1** (highlighted in blue) to **answer the reviewer’s question about when  ECE is preferred over Brier score**.  We illustrate that ECE is preferred when the down-stream decision making has risk preferences. Intuitively, this is because small ECE ensures that the risk, encoded in the forecasted probabilities, is more accurately captured, leading to better *risk-adjusted* decision making – this is the case even when the forecaster has a worse Brier score. We added a new **Appendix Section C.4**, describing a concrete example which we believe clearly highlights this advantage of ECE over Brier score.
> > >
> > > Finally, we changed the “trustworthiness” interpretation of calibration to “reliability”, which is the more classic terminology to refer to calibration error. This dates back to classic statistics work [Murphy 1973], and is also widely used in modern ML (e.g., [Guo et al. 2017] uses “reliability diagram” to describe calibration errors).
> > >
> > > ## **Adding benchmark methodology to main paper**
> > >
> > > We thank the reviewer for highlighting the importance of clearly presenting the benchmark construction in the main paper. After extensive internal discussion, we also agree that a transparent and well-structured description of Prophet Arena’s methodology is essential for clarity, reproducibility, and ease of interpretation. In response, we have substantially revised the paper’s organization: we now provide a complete walkthrough of the benchmark pipeline in (now significantly re-written) **Section 2**, while streamlining the discussion of evaluation metrics to focus on the new insights our framework offers beyond prior work. Please check out the updated PDF draft for the details.

---

> > > > ### Author Response · Authors · 2025-11-25
> > > > **Followup with reviewer reZn**
> > > >
> > > > ## **Topic distribution for events under evaluation**
> > > >
> > > > Below is the detailed category distribution of events collected from Prophet Arena. We have now also updated our paper’s main body (**Third Paragraph of Section 3**, around Line 220) to include a description of these updates.
> > > >
> > > > Given the 810 resolved events up to 2025-09-10 used in our paper, we retain the top categories with the most number of events (Sports, Entertainment, and Politics) and merge other categories into “Other”.The distribution is approximately 81.37% Sports, 5.76% Entertainment, 4.66% Politics, and 8.21% Other.
> > > >
> > > >
> > > > A potential wrinkle might be that this original event distribution is skewed towards sports. To test robustness of our results, we further evaluated all models on two new rebalanced evaluation sets, constructed as follows:
> > > >
> > > > - In **Table 1**, we enforce an even split across categories, i.e., 25% each for Sports, Entertainment, Politics, and Other, with 152 events in total.
> > > >
> > > > - In **Table 2**, we still allow Sports category to take a higher proportion, and adjust to the target distribution {Sports: 50%, Entertainment: 15%, Politics: 15%, Other: 20%}, with 250 events in total. The relative sizes of the three non-Sports categories are kept similar to the original distribution.
> > > >
> > > > We then compare the Brier score across the original dataset and these two rebalanced subsets.
> > > >
> > > >
> > > > **Table 1 (N = 152 events)**
> > > >
> > > > | LLM                                  | ↓ Brier (95% CI)      | Rank |
> > > > |--------------------------------------|------------------------|------|
> > > > | GPT-5ᴿ                               | 0.128 (± 0.015)       | ①    |
> > > > | Grok 4ᴿ                              | 0.141 (± 0.018)       | ②    |
> > > > | Claude Sonnet 4ᴿ (Thinking)         | 0.148 (± 0.020)       | ③    |
> > > > | Gemini 2.5 Flashᴿ (Reasoning)        | 0.157 (± 0.021)       | ④    |
> > > > | Llama 4 Scout                        | 0.210 (± 0.025)       | ⑤    |
> > > > | **Market Baseline**                  | 0.149 (± 0.013)       | N/A  |
> > > >
> > > >
> > > >
> > > > ---
> > > >
> > > > **Table 2 (N = 250 events)**
> > > >
> > > > | LLM                                  | ↓ Brier (95% CI)      | Rank |
> > > > |--------------------------------------|------------------------|------|
> > > > | GPT-5ᴿ                               | 0.146 (± 0.013)       | ①    |
> > > > | Grok 4ᴿ                              | 0.156 (± 0.014)       | ②    |
> > > > | Claude Sonnet 4ᴿ (Thinking)         | 0.162 (± 0.016)       | ③    |
> > > > | Gemini 2.5 Flashᴿ (Reasoning)        | 0.165 (± 0.016)       | ④    |
> > > > | Llama 4 Scout                        | 0.202 (± 0.019)       | ⑤    |
> > > > | **Market Baseline**                  | 0.164 (± 0.010)       | N/A  |
> > > >
> > > > ---
> > > >
> > > > Across the original (**Table 2** in our paper) and both rebalanced distributions, the relative rankings of the LLM predictors remain remarkably stable. When the proportion of Sports events is reduced, all models exhibit moderately improved Brier scores, suggesting that Sports events are relatively more challenging to forecast. Crucially, however, the performance gaps and the comparative ordering between models are preserved. This shows that despite our current benchmark being skewed toward Sports, our evaluation of predictive intelligence remains relevant and consistent in general.
> > > >
> > > > Regarding event selection, Prophet Arena draws forecasting questions directly from prediction markets, using criteria such as popularity, trading volume, and liquidity. Because the benchmark is live and event-driven, temporal shifts in market activity (e.g., surges in Sports or U.S. politics) naturally influence the topic mix. By anchoring on high-volume, liquid markets, we intentionally evaluate LLMs on events where humans have strong financial incentives to gather information and trade, and where prices are already used in practice as probabilistic forecasts. This provides us a strong market baseline: if an LLM can match or outperform the market on these contracts, it provides strong evidence of predictive intelligence; such comparisons are far more informative than evaluating on arbitrary or low-interest questions with little real-world signal.
> > > >
> > > > To further address potential distributional shifts, we are actively developing tools for reweighting and subsampling events by topic category. Since the original submission, we have also begun integrating Polymarket as an additional data source, substantially broadening coverage in politics, economics, entertainment, and other non-sports domains. As we report in the newly added **Appendix C.8**, the current rebalancing experiments already show that the present market skew does not meaningfully alter the core comparative findings. We have clarified these insights in the revised draft and plan to release topic-balanced evaluation views in future public versions of the benchmark.

---

> > > > > ### Comment · Reviewer_reZn · 2025-11-26
> > > > >
> > > > > Thanks for addressing my concerns :)
> > > > >
> > > > > I have increased the score.

---

> > > > > > ### Author Response · Authors · 2025-11-26
> > > > > >
> > > > > > We would like to sincerely thank the reviewer for the time, insightful comments and super constructive suggestions. They really helped improve the soundness and presentations of our work significantly.

---

> > > > > > > ### Comment · Reviewer_reZn · 2025-11-26
> > > > > > >
> > > > > > > I'm glad it helped. Especially appreciate the effort on testing the models under different data distributions and showing rankings stay robust. It anticipated and mitigated an important problem.

---

### Author Response · Authors · 2025-11-22
**Global Response**

We thank all reviewers for their thoughtful and constructive feedback. **We have added several new details to the revised PDF submission (highlighted in blue), and we reference these updated sections in our responses below. Below, we provide a consolidated response to address some of the questions and concerns raised across multiple reviews.**

## **New Experiments on Search**

Since the initial submission, we have continued to work on Prophet Arena and have added three new searchers in addition to gpt-4o: Gemini 2.5 Pro, Grok 4 Fast, and o3. During prediction, all the searchers are deployed separately at the same time. Each predictor will then make a prediction based solely on a single searcher’s collected sources. During this process, every prediction model will only have access to sources from a single searcher. To compare searchers’ performance, we compute the average Brier score of all the predictions that used this searcher’s sources. We report the average Brier score of each searcher below, evaluated on ~7000 predictions collected for each searcher:

| Searcher          |  Average Brier score |
|----------------|---------------|
| gemini-2.5-pro   | 0.232  |
| grok-4-fast   | 0.228    |
| o3     | 0.185   |
| gpt-4o | 0.192 |

We observe that o3 achieves the best performance. However, in practice, o3 occasionally fails to return a result due to its slower response time or unstable endpoint and is also substantially more expensive to run at scale. While gemini-2.5-pro and grok-4-fast are cheaper and have strong reasoning capabilities, our manual analysis indicates that they hallucinate or partially fabricate sources noticeably more often, explaining their worse Brier scores. Therefore, gpt-4o remains the most practical and competitive representative searcher tested. We do, however, note that Prophet Arena is model-agnostic and plan to continue monitoring and/or incorporating new or improved searchers as they become available.

**Does varying the searcher significantly change model rankings?** (in response to reviewer reZn):

In the table below, we report the Brier score ranking of each predictor by searcher (averaged over ~7000 predictions for each searcher). Each row corresponds to a predictor, each column to a searcher, and each cell gives that predictor’s rank when considering only its predictions that used the corresponding searcher’s sources (1 = best Brier score among such predictors, larger values = worse). We observe that OpenAI predictors generally rank within the top-5 not only when using gpt-4o’s retrieved context, but also when using context from other searchers such as google/gemini-2.5-pro and x-ai/grok-4-fast.

Notably, for the o3 search column the OpenAI predictors are not the top-ranked models as google/gemini-2.5-pro and x-ai/grok-4 achieve better rankings there. This indicates that matching the searcher and predictor family does not automatically confer an advantage. Overall, this pattern suggests that the strong performance of OpenAI models cannot be solely attributed to gpt-4o search yielding “in-distribution” context, as they remain competitive (and sometimes underperform) even when conditioned on contexts retrieved by non-OpenAI searchers.

| predictor \ searcher   | google/gemini-2.5-pro | gpt-4o | o3 | x-ai/grok-4-fast |
|------------------------|-----------------------|--------|----|------------------|
| google/gemini-2.5-pro  | 13                    | 8      | 3  | 7                |
| gpt-5-high             | 2                     | 3      | 9  | 1                |
| gpt-5-minimal          | 1                     | 5      | 6  | 4                |
| o3                     | 5                     | 2      | 4  | 3                |
| x-ai/grok-4            | 7                     | 4      | 2  | 5                |



## **Additional Experiments to Validate LLM-as-judge using Human Raters**


To validate our LLM-as-a-judge approach, we conducted a blinded human adjudication study with 170 randomly sampled events, where human raters independently scored reasoning traces across all five rubric dimensions without knowledge of model identity or LLM scores. We found strong concordance between human and LLM ratings, with average disagreements of < 0.5 points (out of a maximum of 4 difference points) and the vast majority (>94%) of ratings agreeing within a single point across all dimensions. Full details of the validation study are provided in Appendix D.7.1. Summary statistics for these human–LLM score differences are reported below.

| Rubric Category               	| Mean Absolute Difference (0-4 scale) (Variance) |
|-------------------------------------- |---------------------------------------|
| Sources Used                   	| 0.43 (0.30)			|
| Evidence Extracted          	| 0.42 (0.27)			|
| Combination Weighting  	| 0.43 (0.36)			|
| Uncertainties Counterpoints | 0.44 (0.28)			|
| Mapping To Final Probs.	| 0.37 (0.29)			|

---

### Meta-Review · Area_Chair_mdeh · 2026-01-05

**Summary:**

This paper introduces a novel evaluation setting for large language models by studying their ability to predict future real-world events using data sourced from online prediction markets. The authors formulate this task as a new benchmark, Prophet Arena, and evaluate a wide range of language models on probabilistic forecasting performance. The idea is original and compelling, directly addressing a question that naturally attracts broad interest: whether language models can outperform, match, or lag behind human collective intelligence in forecasting future outcomes.

Among the four reviewers, three provided positive initial scores, while one reviewer initially recommended rejection. Through detailed discussion and iterative clarification with the authors, this reviewer gradually raised their score, first to 4 and ultimately to 6, which is a positive recommendation. Overall, the review trajectory reflects increasing consensus toward the value of the work.

As the Area Chair, I read the paper, all reviewer comments, the authors’ rebuttal, and the subsequent discussion between reviewers and authors. Reaching a final decision for this paper is non-trivial, as it sits in a particularly delicate position. On one hand, the topic is novel, and the benchmark design captures an aspect of model capability that is both practically relevant and intellectually intriguing. On the other hand, there remain important conceptual questions that are not fully resolved in the current manuscript.

**Reviewer Concerns:**

One reviewer focused primarily on issues of reproducibility, interpretability, and positioning. Their comments questioned how transparent and fair the overall evaluation pipeline is, how search and retrieval choices might influence outcomes, and whether the benchmark is clearly positioned relative to existing forecasting literature. Through discussion, many of these concerns were partially addressed by additional analyses and clarifications, leading this reviewer to maintain a positive score.

Another reviewer raised concerns related to robustness, including dependence on a particular search or retrieval mechanism, sensitivity to event distributions, and the behavior of models near event resolution. The authors responded with additional experiments and analyses, which helped stabilize the reviewer’s assessment and resulted in a consistently positive evaluation.

The reviewer who was initially most critical raised deeper conceptual issues, including whether the benchmark clearly measures well-defined cognitive abilities, whether forecasting is a sufficiently grounded proxy for intelligence, and whether prediction markets constitute a stable and appropriate baseline. Through extended discussion, clarification, and additional evidence, this reviewer acknowledged improvements and ultimately raised their score to a positive level, while still noting that some conceptual questions remain open.

**Reviewer Scores:**

During the discussion, the majority of reviewers explicitly indicated their intention to revise their scores upward.

Beyond summarizing the reviewers’ discussions, I would like to add several Area Chair level observations.

While the forecasting task itself is highly interesting, the paper does not sufficiently emphasize or analyze the fundamental predictability of the task. The manuscript reports prediction accuracy, calibration metrics, and comparisons across models, but it remains unclear what these numbers ultimately signify in terms of task difficulty and information structure. For example, when two models are given the same information and produce different forecasts, the paper does not deeply examine why these differences arise or what evidence or reasoning pathways drive them.

In particular, for domains such as sports betting and event forecasting, an important open question is what signals actually make an event predictable. The current paper does not clearly articulate whether the benchmark primarily evaluates retrieval ability, reasoning over noisy evidence, calibration under uncertainty, or some combination thereof. While these questions are partially touched upon in analysis sections, they are not sufficiently foregrounded in the main narrative. Given how novel and conceptually rich this task is, a deeper discussion of what exactly is being measured would significantly strengthen the contribution.

For this reason, the final decision is difficult. The paper is not flawless, and there remain important open questions about task interpretation and capability attribution. However, given that multiple reviewers ultimately expressed positive recommendations, and considering the open and transparent nature of the OpenReview process in which this Meta Review will be recorded, I believe the paper merits acceptance.

That said, I strongly encourage the authors to further revise the writing to more explicitly address the above concerns, particularly by clarifying the nature of predictability in the task and more clearly articulating what aspects of language model capability are being evaluated. Strengthening this discussion would substantially improve the clarity, impact, and long-term value of the benchmark.

---

### Decision · Program_Chairs · 2026-01-26

Accept (Poster)